# (Nearly) Optimal Algorithms for Private Online Learning in Full-information and Bandit Settings

**Adam Smith**[*]
Pennsylvania State University
asmith@cse.psu.edu

**Abhradeep Thakurta**[†]
Stanford University and
Microsoft Research Silicon Valley Campus
b-abhrag@microsoft.com

## Abstract

We give differentially private algorithms for a large class of online learning algorithms, in both the full information and bandit settings. Our algorithms aim to minimize a *convex* loss function which is a sum of smaller convex loss terms, one for each data point. To design our algorithms, we modify the popular *mirror descent* approach, or rather a variant called *follow the approximate leader*.

The technique leads to the first nonprivate algorithms for private online learning in the bandit setting. In the full information setting, our algorithms improve over the regret bounds of previous work (due to Dwork, Naor, Pitassi and Rothblum (2010) and Jain, Kothari and Thakurta (2012)). In many cases, our algorithms (in both settings) match the dependence on the input length, $T$, of the optimal nonprivate regret bounds up to logarithmic factors in $T$. Our algorithms require logarithmic space and update time.

## 1  Introduction

This paper looks at the information leaked by online learning algorithms, and seeks to design accurate learning algorithms with rigorous privacy guarantees – that is, algorithms that provably leak very little about individual inputs.

Even the output of offline (batch) learning algorithms can leak private information. The dual form of a support vector machine's solution, for example, is described in terms of a small number of exact data points, revealing these individuals' data in the clear. Considerable effort has been devoted to designing batch learning algorithms satisfying *differential privacy* (a rigorous notion of privacy that emerged from the cryptography literature [DMNS06, Dwo06]), for example [BDMN05, KLN+08, CM08, CMS11, Smi11, KST12, JT13, DJW13].

In this work we provide a general technique for making a large class of online learning algorithms differentially private, in both the full information and bandit settings. Our technique applies to algorithms that aim to minimize a *convex* loss function which is a sum of smaller convex loss terms, one for each data point. We modify the popular *mirror descent* approach (or rather a variant called *follow the approximate leader*) [Sha11, HAK07].

In most cases, the modified algorithms provide similar accuracy guarantees to their nonprivate counterparts, with a small (logarithmic in the stream length) blowup in space and time complexity.

**Online (Convex) Learning:** We begin with the *full information* setting. Consider an algorithm that receives a stream of inputs $F = \langle f_1, ...., f_T \rangle$, each corresponding to one individual's data. We interpret each input as a loss function on a parameter space $\mathcal{C}$ (for example, it might be one term

---

[*]Supported by NSF awards #0941553 and #0747294.

[†]Supported by Sloan Foundation fellowship and Microsoft Research.

in a convex program such as the one for logistic regression). The algorithm's goal is to output a sequence of parameter estimates $w_1, w_2, ...$, with each $w_t$ in $\mathcal{C}$, that roughly minimizes the errors $\sum_t f_t(w_t)$. The difficulty for the algorithm is that it computes $w_t$ based only on $f_1, ..., f_{t-1}$. We seek to minimize the *a posteriori regret*,

$$Regret(T) = \sum_{t=1}^{T} f_t(w_t) - \min_{w \in \mathcal{C}} \sum_{t=1}^{T} f_t(w) \tag{1}$$

In the *bandit* setting, the input to the algorithms consists only of $f_1(w_1), f_2(w_2), ...$. That is, at each time step $t$, the algorithm learns only the cost $f_{t-1}(w_{t-1})$ of the choice $w_{t-1}$ it made at the previous time step, rather than the full cost function $f_{t-1}$.

We consider three types of adversarial input selection: An *oblivious* adversary selects the input stream $f_1, ..., f_T$ ahead of time, based on knowledge of the algorithm but not of the algorithm's random coins. A *(strongly) adaptive* adversary selects $f_t$ based on the output so far $w_1, w_2, ..., w_t$ (but not on the algorithm's internal random coins).

Both the full-information and bandit settings are extensively studied in the literature (see, e.g., [Sha11, BCB12] for recent surveys). Most of this effort has been spent on online learning problems are *convex*, meaning that the loss functions $f_t$ are convex (in $w$) and the parameter set $\mathcal{C} \subseteq \mathbb{R}^p$ is a convex set (note that one can typically "convexify" the parameter space by randomization). The problem dimension $p$ is the dimension of the ambient space containing $\mathcal{C}$.

We consider various restrictions on the cost functions, such as Lipschitz continuity and strong convexity. A function $f : \mathcal{C} \to \mathbb{R}$ is $L$-Lipschitz with respect to the $\ell_2$ metric if $|f(x) - f(y)| \leq L\|x - y\|_2$ for all $x, y \in \mathcal{C}$. Equivalently, for every $x \in \mathcal{C}^0$ (the interior of $\mathcal{C}$) and every subgradient $z \in \partial f(x)$, we have $\|z\|_2 \leq L$. (Recall that $z$ is a subgradient of $f$ at $x$ if the function $\tilde{f}(y) = f(x) + \langle z, y - x \rangle$ is a lower bound for $f$ on all of $\mathcal{C}$. If $f$ is convex, then a subgradient exists at every point, and the subgradient is unique if and only if $f$ is differentiable at that point.) The function $f$ is $H$-strongly convex w.r.t. $\ell_2$ if for every $y \in \mathcal{C}$, we can bound $f$ below on $\mathcal{C}$ by a quadratic function of the form $\tilde{f}(y) = f(x) + \langle z, y - x \rangle + \frac{H}{2}\|y - x\|_2^2$. If $f$ is twice differentiable, $H$-strong convexity is equivalent to the requirement that all eigenvalues of $\nabla^2 f(w)$ be at least $H$ for all $w \in \mathcal{C}$.

We denote by $\mathcal{D}$ the set of allowable cost functions; the input sequence thus lies in $\mathcal{D}^T$.

**Differential Privacy, and Challenges for Privacy in the Online Setting:** We seek to design online learning algorithms that satisfy *differential privacy* [DMNS06, Dwo06], which ensures that the amount of information an adversary learns about a particular cost function $f_t$ in the function sequence $F$ is almost independent of its presence or absence in $F$. Each $f_t$ can be thought as private information belonging to an individual. The appropriate notion of privacy here is when the entire sequence of outputs of the algorithms $(\hat{w}_1, ..., \hat{w}_T)$ is revealed to an attacker (the *continual observation* setting [DNPR10]). Formally, we say two input sequences $F, F' \in \mathcal{D}^T$ are *neighbors* if they differ only in one entry (say, replacing $f_t$ by $f_t'$).

**Definition 2** (Differential privacy [DMNS06, Dwo06, DNPR10])**.** *A randomized algorithm $\mathcal{A}$ is $(\epsilon, \delta)$-differentially private if for every two neighboring sequences $F, F' \in \mathcal{D}^T$, and for every event $\mathcal{O}$ in the output space $\mathcal{C}^T$,*

$$\Pr[\mathcal{A}(F) \in \mathcal{O}] \leq e^{\epsilon} \Pr[\mathcal{A}(F') \in \mathcal{O}] + \delta. \tag{2}$$

*If $\delta$ is zero, then we simply say $\mathcal{A}$ is $\epsilon$-differentially private.*

Here $\mathcal{A}(F)$ refers to the entire sequence of outputs produced by the algorithm during its execution.[1] Our protocols all satisfy $\epsilon$-differential privacy (that is, with $\delta = 0$). We include $\delta$ in the definition for comparison with previous work.

Differential privacy provides meaningful guarantees in against an attacker who has access to considerable side information: the attacker learns the same things about someone whether or not their data were actually used (see [KS08, DN10, KM12] for further discussion).

Differential privacy is particularly challenging to analyze for online learning algorithms, since a change in a single input at the beginning of the sequence may affect outputs at all future times in ways that are hard to predict. For example, a popular algorithm for online learning is *online gradient descent*: at each time step, the parameter is updated as $w_{t+1} = \Pi_{\mathcal{C}}(w_{t-1} - \eta_t \nabla f_{t-1}(w_{t-1}))$, where $\Pi_{\mathcal{C}}(x)$ the nearest point to $x$ in $\mathcal{C}$, and $\eta_t > 0$ is a parameter called the learning rate. A change in an input $f_i$ (replacing it with $f_i'$) leads to changes in all subsequent outputs $w_{i+1}, w_{i+2}, ...$, roughly pushing them in the direction of $\nabla f_i(w_i) - \nabla f_i'(w_i)$. The effect is amplified by the fact that the gradient of subsequent functions $f_{i+1}, f_{i+2}, ...$ will be evaluated at different points in the two streams.

**Previous Approaches:** Despite the challenges, there are several results on differentially private online learning. A special case, "learning from experts" in the full information setting, was discussed in the seminal paper of Dwork, Naor, Pitassi and Rothblum [DNPR10] on privacy under continual observation. In this case, the set of available actions is the simplex $\Delta(\{1, ..., p\})$ and the functions $f_i$ are linear with coefficients in $\{0, 1\}$ (that is, $f_t(w) = \langle w, c_t \rangle$ where $c_t \in \{0, 1\}^p$). Their algorithm guarantees a weaker notion of privacy than the one we consider[2] but, when adapted to our stronger setting, it yields a regret bound of $O(p\sqrt{T}/\epsilon)$.

Jain, Kothari and Thakurta [JKT12] defined the general problem of private online learning, and gave algorithms for learning convex functions over convex domains in the full information setting. They gave algorithms that satisfy $(\epsilon, \delta)$-differential privacy with $\delta > 0$ (our algorithms satisfy the stronger variant with $\delta = 0$). Specifically, their algorithms have regret $\tilde{O}(\sqrt{T}\log(1/\delta)/\epsilon)$ for Lipshitz-bounded, strongly convex cost functions and $\tilde{O}(T^{2/3}\log(1/\delta)/\epsilon)$ for general Lipshitz convex costs. The idea of [JKT12] for learning strongly convex functions is to bound the sensitivity of the entire vector of outputs $w_1, w_2, ...$ to a change in one input (roughly, they show that when $f_i$ is changed, a subsequent output $w_j$ changes by $O(1/|j - i|)$).

Unfortunately, the regret bounds obtained by previous work remain far from the best nonprivate bounds. [Zin03] gave an algorithm with regret $O(\sqrt{T})$ for general Lipshitz functions, assuming $L$ and the diameter $\|\mathcal{C}\|_2$ of $\mathcal{C}$ are constants. $\Omega(\sqrt{T})$ regret is necessary (see, e.g., [HAK07]), so the dependence on $T$ of [Zin03] is tight. When cost functions in $F$ are $H$-strongly convex for constant $H$, then the regret can be improved to $O(\log T)$ [HAK07], which is also tight. In this work, we give new algorithms that match these nonprivate bounds' dependence on $T$, up to $(\mathrm{poly}\log T)/\epsilon$ factors.

We note that [JKT12] give one algorithm for a specific strongly convex problem, online linear regression, with regret $\mathrm{poly}(\log T)$. One can view that algorithm as a special case of our results.

We are not aware of any previous work on privacy in the bandit setting. One might expect that bandit learning algorithms are *easier* to make private, since they access data in a much more limited way. However, even nonprivate algorithms for bandit learning are very delicate, and private versions had until now proved elusive.

**Our Results:** In this work we provide a technique for making a large class of online learning algorithms differentially private, in both the full information and bandit settings. In both cases, the idea is to search for algorithms whose decisions at time $t$ depend only on previous time steps through a *sum* of observations made at times $1, 2, ..., t$. Specifically, our algorithms work by measuring the gradient $\nabla f_t(w_t)$ when $f_t$ is learned, and maintaining a differentially private running sum of the gradients observed so far. We maintain this sum using the tree-based sum protocol of [DNPR10, CSS10]. We then show that a class of learning algorithms known collectively as *follow the approximate leader* (the version we use is due to [HAK07]) can be run given only these noisy sums, and that their regret can be bounded even when these sums are inaccurate.

Our algorithms can be run with space $O(\log T)$, and require $O(\log T)$ running time at each step.

Our contributions for the full information setting and their relation to previous work is summarized in Table 1. Our main algorithm, for strongly convex functions, achieves regret $O(\frac{\log^{2.5} T}{\epsilon})$, ignoring factors of the dimension $p$, Lipschitz continuity $L$ and strong convexity $H$. When strong convexity is not guaranteed, we use regularization to ensure it (similar to what is done in nonprivate settings, e.g. [Sha11]). Setting parameters carefully, we get regret of $O(\sqrt{\frac{T \log^{2.5} T}{\epsilon}})$. These bounds essentially match the nonprivate lower bounds of $\Omega(\log T)$ and $\Omega(\sqrt{T})$, respectively.

The results in the full information setting apply even when the input stream is chosen adaptively as a function of the algorithm's choices at previous time steps. In the bandit setting, we distinguish between oblivious and adaptive adversaries.

Furthermore, in the bandit setting, we assume that $\mathcal{C}$ is sandwiched between two concentric $L_2$-balls of radii $r$ and $R$ (where $r < R$). We also assume that for all $w \in \mathcal{C}$, $|f_t(w)| \leq B$ for all $t \in [T]$. Similar assumption were made in [FKM05, ADX10].

Our results are summarized in Table 2. For most of the settings we consider, we match the dependence on $T$ of the best nonprivate algorithm, though generally not the dependence on the dimension $p$.

| Function class | Previous private upper bound. | Our algorithm | Nonprivate lower bound |
|---|---|---|---|
| Learning with experts (linear functions over $\mathcal{C} = \Delta(\{1,...,p\})$ | $\tilde{O}(p\sqrt{T}/\epsilon)$ [DNPR10] | $O(\sqrt{pT \log^{2.5} T}/\epsilon)$ | $\Omega(\sqrt{T \log p})$ |
| Lipshitz | $\tilde{O}(\sqrt{p}T^{2/3} \log(1/\delta)/\epsilon)$ [JKT12] | $O(\sqrt{pT \log^{2.5} T}/\epsilon)$ | $\Omega(\sqrt{T})$ |
| Lipshitz and strongly convex | $\tilde{O}(\sqrt{pT} \log^2(1/\delta)/\epsilon)$ [JKT12] | $O(p \log^{2.5} T/\epsilon)$ | $\Omega(\log T)$ |

Table 1: Regret bounds for online learning in the full information setting. Bounds in lines 2 and 3 hide the (polynomial) dependencies on parameters $L, H$. Notation $\tilde{O}(\cdot)$ hides $poly(\log(T))$ factors.

| Function class | Our result | Best nonprivate bound |
|---|---|---|
| Learning with experts (linear functions over $\mathcal{C} = \Delta(\{1,...,p\})$ | $\tilde{O}(pT^{3/4}/\epsilon)$ | $O(\sqrt{T})$ [AHR08] |
| Lipshitz | $\tilde{O}(pT^{3/4}/\epsilon)$ | $O(pT^{3/4})$ [FKM05] |
| Lipshitz and strongly convex (Adaptive) | $\tilde{O}(pT^{3/4}/\epsilon)$ | $O(p^{2/3}T^{3/4})$[ADX10] |
| Lipshitz and strongly convex (Oblivious) | $\tilde{O}(pT^{2/3}/\epsilon)$ | $O(p^{2/3}T^{2/3})$[ADX10] |

Table 2: Regret bounds for online learning in the bandit setting. In all these settings, the best known nonprivate *lower* bound is $\sqrt{T}$. The $\tilde{O}(\cdot)$ notation hides $poly$ log factors in $T$. Bounds hide polynomial dependencies on $L, H, r$ and $R$.

In the remainder of the text, we refer to appendices for many of the details of algorithms and proofs. The appendices can be found in the "Supplementary Materials" associated to this paper.

## 2    Private Online Learning: Full-information Setting

In this section we adapt the *Follow The Approximate Leader* (FTAL) algorithm of [HAK07] to design a differentially private variant. Our modified algorithm, which we call *Private Follow The*

*Approximate Leader* (PFTAL), needs a new regret analysis as we have to deal with randomness due to differential privacy.

## 2.1 Private Follow The Approximate Leader (PFTAL) with Strongly Convex Costs

---

**Algorithm 1** Differentially Private Follow the Approximate Leader (PFTAL)

---

**Input:** Cost functions: $\langle f_1, \cdots, f_T \rangle$ (in an online sequence), strong convexity parameter: $H$, Lipschitz constant: $L$, convex set: $\mathcal{C} \subseteq \mathbb{R}^p$ and privacy parameter: $\epsilon$.

1: $\hat{w}_1 \leftarrow$ Any vector from $\mathcal{C}$. **Output** $\hat{w}_1$.
2: Pass $\bigtriangledown f_1(\hat{w}_1)$, $L_2$-bound $L$ and privacy parameter $\epsilon$ to the *tree based aggregation protocol* and receive the current partial sum in $\hat{v}_1$.
3: **for** time steps $t \in \{1, \cdots, T-1\}$ **do**
4: $\quad \hat{w}_{t+1} \leftarrow \arg\min\limits_{w \in \mathcal{C}} \langle \hat{v}_t, w \rangle + \frac{H}{2} \sum\limits_{\tau=1}^{t} \|w - \hat{w}_\tau\|_2^2$. **Output** $\hat{w}_t$.
5: $\quad$ Pass $\bigtriangledown f_{t+1}(\hat{w}_{t+1})$, $L_2$-bound $L$ and privacy parameter $\epsilon$ to the *tree-based protocol* (Algorithm 2) and receive the current partial sum in $\hat{v}_{t+1}$.
6: **end for**

---

The main idea in PFTAL algorithm is to execute the well-known Follow The Leader algorithm (FTL) algorithm [Han57] using quadratic approximations $\tilde{f}_1, \cdots, \tilde{f}_T$ of the cost functions $f_1, \cdots, f_T$. Roughly, at every time step $(t+1)$, PFTAL outputs a vector $w$ that approximately minimizes the sum of the approximations $\tilde{f}_1, \cdots, \tilde{f}_t$ over the convex set $\mathcal{C}$.

Let $\hat{w}_1, \cdots, \hat{w}_t$ be the sequence of outputs produced in the first $t$ time steps, and let $f_t$ be the cost-function at step $t$. Consider the following quadratic approximation to $f_t$ (as in [HAK07]). Define

$$\tilde{f}_t(w) = f_t(\hat{w}_t) + \langle \bigtriangledown f_t(\hat{w}_t), w - \hat{w}_t \rangle + \frac{H}{2} \|w - \hat{w}_t\|_2^2 \tag{3}$$

where $H$ is the strong convexity parameter. Notice that $f_t$ and $\tilde{f}_t$ have the same value and gradient at $\hat{w}_t$ (that is, $f_t(\hat{w}_t) = \tilde{f}_t(\hat{w}_t)$ and $\bigtriangledown f_t(\hat{w}_t) = \bigtriangledown \tilde{f}_t(\hat{w}_t)$). Moreover, $\tilde{f}_t$ is a lower bound for $f_t$ everywhere on $\mathcal{C}$.

Let $\tilde{w}_{t+1} = \arg\min\limits_{w \in \mathcal{C}} \sum\limits_{\tau=1}^{t} \tilde{f}_\tau(w)$ be the "leader" corresponding to the cost functions $\tilde{f}_1, \cdots, \tilde{f}_t$. Minimizing the sum of $\tilde{f}_t(w)$ is the same as minimizing the sum of $\tilde{f}_t(w) - f_t(\hat{w}_t)$, since subtracting a constant term won't change the minimizer. We can thus write $\tilde{w}_{t+1}$ as

$$\tilde{w}_{t+1} = \arg\min\limits_{w \in \mathcal{C}} \langle \sum\limits_{\tau=1}^{t} \bigtriangledown f_t(\hat{w}_\tau), w \rangle + \frac{H}{2} \sum\limits_{\tau=1}^{t} \|w - \hat{w}_\tau\|_2^2 \tag{4}$$

Suppose, $\hat{w}_1, \cdots, \hat{w}_t$ have been released so far. To release a private approximation to $\tilde{w}_{t+1}$, it suffices to approximate $v_{t+1} = \sum_{\tau=1}^{t} \bigtriangledown f_t(\hat{w}_\tau)$ while ensuring differential privacy. If we fix the previously released information $\hat{w}_\tau$, then changing any one cost function will only change one of the summands in $v_{t+1}$.

With the above observation, we abstract out the following problem: Given a set of vectors $z_1, \cdots, z_T \in \mathbb{R}^p$, compute all the partial sums $v_t = \sum\limits_{\tau=1}^{t} z_\tau$, while preserving privacy. This problem is well studied in the privacy literature. Assuming each $z_t$ has $L_2$-norm of at most $L'$, the following *tree-based aggregation* scheme will ensure that in expectation, the noise (in terms of $L_2$-error) in each of $v_t$ is $O\left(pL' \log^{1.5} T/\epsilon\right)$ and the whole sequence $v_1, \cdots, v_T$ is $\epsilon$-differentially private. We now describe the tree-based scheme.

**Tree-based Aggregation [DNPR10, CSS10]:** Consider a complete binary tree. The leaf nodes are the vectors $z_1, \cdots, z_T$. (For the ease of exposition, assume $T$ to be a power of two. In general, we can work with the smallest power of two greater than $T$). Each internal node in the tree stores the sum of all the leaves in its sub-tree. In a differentially private version of this tree, we ensure that each node's sub-tree sum is $(\epsilon/\log_2 T)$-differentially private, by adding a noise vector $b \in \mathbb{R}^p$

whose $L_2$-norm is Gamma distributed and has standard deviation $O(\frac{\sqrt{p}L'\log T}{\epsilon})$. Since each $z_t$ only affects $\log_2 T$ nodes in the tree, by the *composition property* [DMNS06], the complete tree will be $\epsilon$-differentially private. Moreover, the algorithm's error in estimating any partial sum $v_t = \sum_{\tau=1}^{t} z_\tau$ grows as $O(\frac{\sqrt{p}L'\log^2 T}{\epsilon})$, since one can compute $v_t$ from at most $\log T$ nodes in the tree. A formal description of the tree based aggregation scheme in given in Appendix A.

Now we complete the PFTAL algorithm by computing the private version $\hat{w}_{t+1}$ of $\tilde{w}_{t+1}$ in (4) as the minimizer of the perturbed loss function:

$$\hat{w}_{t+1} = \arg\min_{w \in \mathcal{C}} \langle \hat{v}_t, w \rangle + \frac{H}{2} \sum_{\tau=1}^{t} \|w - \hat{w}_\tau\|_2^2 \qquad (5)$$

Here $\hat{v}_t$ is the noisy version of $v_t$, computed using the tree-based aggregation scheme. A formal description of the algorithm is given in Algorithm 1.

**Note on space complexity:** For simplicity, in the description of tree based aggregation scheme (Algorithm 2 in Appendix A) we maintain the complete binary tree. However, it is not hard to show at any time step $t$, it suffices to keep track of the vectors (of partial sums) in the path from $z_t$ to the root of the tree. So, the amount of space required by the algorithm is $O(\log T)$.

### 2.1.1 Privacy and Utility Guarantees for PFTAL (Algorithm 1)

In this section we provide the privacy and regret guarantees for the PFTAL algorithm (Algorithm 1). For detailed proofs of the theorem statements, see Appendix B.

**Theorem 3** (Privacy guarantee). *Algorithm 1 is $\epsilon$-differentially private.*

*Proof Sketch.* Given the binary tree, the sequence $\hat{w}_2, \cdots, \hat{w}_T$ is completely determined. Hence, it suffices to argue privacy for the collection of noisy sums associated to nodes in the binary tree. At first glance, it seems that each loss function affects only one leaf in the tree, and hence at most $\log T$ of the nodes' partial sums. If it were true, that statement would make the analysis simple. The analysis is delicate, however, since the value (gradient $z_\tau$) at a leaf $\tau$ in the tree depends on the partial sums that are released before time $\tau$. Hence, changing one loss function $f_t$ actually affects *all* subsequent partial sums. One can get around this by using the fact that differential privacy composes adaptively [DMNS06]: we can write the computations done on a particular loss function $f_t$ as a sequence of $\log T$ smaller differentially private computations, where the each computation in the sequence depends on the outcome of previous ones. See Appendix B for details. $\qquad\square$

In terms of regret guarantee, we show that our algorithm enjoys regret of $O(p\log^{2.5} T)$ (assuming other parameters to be constants). Compared to the non-private regret bound of $O(\log T)$, our regret bound has an extra $\log^{1.5} T$ factor and an *explicit* dependence on the dimensionality ($p$). A formal regret bound for PFTAL algorithm is given in Theorem 4.

**Theorem 4** (Regret guarantee). *Let $f_1, \cdots, f_T$ be $L$-Lipschitz, $H$-strongly convex functions and let $\mathcal{C} \subseteq \mathbb{R}^p$ be a fixed convex set. For adaptive adversaries, the expected regret satisfies:*

$$\mathbb{E}\left[Regret(T)\right] = O\left(\frac{p(L + H\|\mathcal{C}\|_2)^2 \log^{2.5} T}{\epsilon H}\right).$$

*Here expectation is taken over the random coins of the algorithm and adversary.*

**Results for Lipschitz Convex Costs:** Our algorithm for strongly convex costs can be adapted to arbitrary Lipschitz convex costs by executing Algorithm 1 on functions $h_t(w) = f_t(w) + \frac{H}{2}\|w\|_2^2$ instead of the $f_t$'s. Setting $H = O(p\log^{2.5} T/(\epsilon\sqrt{T}))$ will give us a regret bound of $\tilde{O}(\sqrt{pT}/\epsilon)$. See Appendix C for details.

## 3 Private Online Learning: Bandit Setting

In this section we adapt the Private Follow the Approximate Leader (PFTAL) from Section 2 to the bandit setting. Existing (nonprivate) bandit algorithms for online convex optimization follow

a generic reduction to the full-information setting [FKM05, ADX10], called the "one-point" (or "one-shot") gradient trick. Our adaptation of PFTAL to the bandit setting also uses this technique. Specifically, to define the quadratic lower bounds to the input cost functions (as in (3)), we replace the exact gradient of $f_t$ at $\hat{w}_t$ with a one-point approximation.

In this section we describe our results for strongly convex costs. Specifically, to define the quadratic lower bounds to the input cost functions (as in (3)), we replace the exact gradient of $f_t$ at $\hat{w}_t$ with a one-point approximation. As in the full information setting, one may obtain regret bounds for general convex functions in the bandit setting by adding a strongly convex regularizer to the cost functions.

**One-point Gradient Estimates [FKM05]:** Suppose one has to estimate the gradient of a function $f : \mathbb{R}^p \to \mathbb{R}$ at a point $w \in \mathbb{R}^p$ via a single query access to $f$. [FKM05] showed that one can approximate $\bigtriangledown f(w)$ by $\frac{p}{\beta} f(w + \beta u)u$, where $\beta > 0$ is a small real parameter and $u$ is a uniformly random vector from the $p$-dimensional unit sphere $\mathbb{S}^{p-1} = \{a \in \mathbb{R}^p : \|a\|_2 = 1\}$. More precisely,

$$\bigtriangledown f(w) = \lim_{\beta \to 0} \mathbb{E}_u \left[ \frac{p}{\beta} f(w + \beta u)u \right].$$

For finite, nonzero values of $\beta$, one can view this technique as estimating the gradient of a smoothed version of $f$. Given $\beta > 0$, define $\hat{f}(w) = \mathbb{E}_{v \sim \mathbb{B}^p} [f(w + \beta v)]$ where $\mathbb{B}^p$ is the unit ball in $\mathbb{R}^p$. That is, $\hat{f} = f * U_{\beta \mathbb{B}^p}$ is the convolution of $f$ with the uniform distribution on the ball $\beta \mathbb{B}^p$ of radius $\beta$. By Stokes' theorem, we have $\mathbb{E}_{u \sim \mathbb{S}^{p-1}} \left[ \frac{p}{\beta} f(w + \beta u)u \right] = \bigtriangledown \hat{f}(w)$.

## 3.1 Follow the Approximate Leader (Bandit version): Non-private Algorithm

Let $\tilde{W} = \langle \tilde{w}_1, \cdots, \tilde{w}_T \rangle$ be a sequence of vectors in $\mathcal{C}$ (the outputs of the algorithm). Corresponding to the smoothed function $\hat{f}_t = f * U_{\beta \mathbb{B}^p}$, we define a quadratic lower bound $\hat{g}_t$:

$$\hat{g}_t(w) = \hat{f}_t(\tilde{w}_t) + \langle \bigtriangledown \hat{f}_t(\tilde{w}_t), w - \tilde{w}_t \rangle + \tfrac{H}{2} \|w - \tilde{w}_t\|_2^2 \tag{6}$$

Notice that $\hat{g}_t$ is a uniform lower bound on $\hat{f}_t$ satisfying $\hat{g}_t(\tilde{w}_t) = \hat{f}_t(\tilde{w}_t)$ and $\bigtriangledown \hat{g}_t(\tilde{w}_t) = \bigtriangledown \hat{f}_t(\tilde{w}_t)$.

To define $\hat{g}_t$, one needs access to $\bigtriangledown \hat{f}_t(\tilde{w}_t)$. As suggested above, we replace the true gradient with the one-point estimate. Consider the following proxy $\tilde{g}_t$ for $\hat{g}_t$:

$$\tilde{g}_t(w) = \underbrace{\hat{f}_t(\tilde{w}_t) - \langle \bigtriangledown \hat{f}_t(\tilde{w}_t), \tilde{w}_t \rangle}_{A} + \langle \frac{p}{\beta} f_t(\tilde{w}_t + \beta u_t)u_t, w \rangle + \frac{H}{2} \|w - \tilde{w}_t\|_2^2 \tag{7}$$

where $u_T$ is drawn uniformly from the unit sphere $\mathbb{S}^{p-1}$. Note that in (7) we replaced the gradient of $\hat{f}_t$ with its one-point approximation only in one of its two occurrences (the inner product with $w$).

We would like to define $\tilde{w}_{t+1}$ as the minimizer of the sum of proxies $\sum_{\tau=1}^{t} \tilde{g}_\tau(w)$. One difficulty remains: because $f_t$ is only assumed to be defined on $\mathcal{C}$, the approximation $\frac{p}{\beta} f_t(\tilde{w}_t + \beta u_t)u_t$ is only defined when $\tilde{w}_t$ is sufficiently far inside $\mathcal{C}$. Recall from the introduction that we assume $\mathcal{C}$ contains $r\mathbb{B}^p$ (the ball of radius $r$). To ensure that we only evaluate $f$ on $\mathcal{C}$, we actually minimize over a smaller set $(1 - \xi)\mathcal{C}$, where $\xi = \frac{\beta}{r}$. We obtain:

$$\tilde{w}_{t+1} = \arg \min_{w \in (1-\xi)\mathcal{C}} \sum_{\tau=1}^{t} \tilde{g}_\tau(w) = \arg \min_{w \in (1-\xi)\mathcal{C}} \langle \sum_{\tau=1}^{t} \left( \frac{p}{\beta} f_t(\tilde{w}_t + \beta u_t)u_t \right), w \rangle + \frac{H}{2} \sum_{\tau=1}^{t} \|w - \tilde{w}_\tau\|_2^2 \tag{8}$$

(We have use the fact that to minimize $\tilde{g}_t$, one can ignore the constant term $A$ in (7).)

We can now state the bandit version of FTAL. At each step $t = 1, ..., T$:

1. Compute $\tilde{w}_{t+1}$ using (8).
2. Output $\hat{w}_t = \tilde{w}_t + \beta u_t$.

Theorem 12 (in Appendix D) gives the precise regret guarantees for this algorithm. For adaptive adversaries the regret is bounded by $\tilde{O}(p^{2/3}T^{3/4})$ and for oblivious adversaries the regret is bounded by $\tilde{O}(p^{2/3}T^{2/3})$.

## 3.2 Follow the Approximate Leader (Bandit version): Private Algorithm

To make the bandit version of FTAL $\epsilon$-differentially private, we replace the value $v_t = \sum_{\tau=1}^{t} \left( \frac{p}{\beta} f_t(w_t^\dagger + \beta u_t) u_t \right)$ with a private approximation $v_t^\dagger$ computed using the tree-based sum protocol. Specifically, at each time step $t$ we output

$$w_{t+1}^\dagger = \arg \min_{w \in (1-\xi)\mathcal{C}} \langle v_t^\dagger, w \rangle + \frac{H}{2} \sum_{\tau=1}^{t} \|w - w_\tau^\dagger\|_2^2. \qquad (9)$$

See Algorithm 3 (Appendix E.1) for details.

**Theorem 5** (Privacy guarantee)**.** *The bandit version of Private Follow The Approximate Leader (Algorithm 3) is $\epsilon$-differentially private.*

The proof of Theorem 5 is exactly the same as of Theorem 3, and hence we omit the details.

In the following theorem we provide the regret guarantee of the Private FTAL (bandit version). For a complete proof, see Appendix E.2.

**Theorem 6** (Regret guarantee)**.** *Let $\mathbb{B}^p$ be the p-dimensional unit ball centered at the origin and $\mathcal{C} \subseteq \mathbb{R}^p$ be a convex set such that $r\mathbb{B}^p \subseteq \mathcal{C} \subseteq R\mathbb{B}^p$ (where $0 < r < R$). Let $f_1, \cdots, f_T$ be L-Lipschitz, H-strongly convex functions such that for all $w \in \mathcal{C}$, $|f_i(w)| \leq B$. Setting $\xi = \beta/r$ in the bandit version of Private Follow The Approximate Leader (Algorithm 3 in Appendix E.1), we obtain the following regret guarantees.*

1. **(Oblivious adversary)** *With $\beta = \frac{p}{T^{1/3}}$, $\mathbb{E}[Regret(T)] \leq \tilde{O}\left(pT^{2/3}\chi\right)$*

2. **(Adaptive adversary)** *With $\beta = \frac{p}{T^{1/4}}$, $\mathbb{E}[Regret(T)] \leq \tilde{O}\left(pT^{3/4}\chi\right)$*

*Here $\chi = \left( BR + (1 + R/r)L + \frac{(H\|\mathcal{C}\|_2 + B)^2}{H}\left(1 + \frac{B}{\epsilon}\right) \right)$. The expectations are taken over the randomness of the algorithm and the adversary.*

One can remove the dependence on $r$ in Thm. 6 by rescaling $\mathcal{C}$ to isotropic position. This increases the expected regret bound by a factor of $(LR + \|\mathcal{C}\|_2)$. See [FKM05] for details.

**Bound for general convex functions:** Our results in this section can be extended to the setting of arbitrary Lipshitz convex costs via regularization, as in Section C (by adding $\frac{H}{2}\|w\|_2^2$ to each cost function $f_t$). With the appropriate choice of $H$ the regret scales as $\tilde{O}(T^{3/4}/\epsilon)$ for both oblivious and adaptive adversaries. See Appendix E.3 for details.

## 4 Open Questions

Our work raises several interesting open questions: First, our regret bounds with general convex functions have the form $\tilde{O}(\sqrt{T}/\epsilon)$. We would like to have a regret bound where the parameter $1/\epsilon$ is factored out with lower order terms in the regret, *i.e.,* we would like to have regret bound of the form $O(\sqrt{T}) + o(\sqrt{T}/\epsilon)$.

Second, our regret bounds for convex bandits are worse than the non-private bounds for linear and multi-arm bandits. For multi-arm bandits [ACBF02] and for linear bandits [AHR08], the non-private regret bound is known to be $O(\sqrt{T})$. If we use our private algorithm in this setting, we will incur a regret of $\tilde{O}(T^{2/3})$. Can we get $O(\sqrt{T})$ regret for multi-arm or linear bandits?

Finally, bandit algorithms require internal randomness to get reasonable regret guarantees. Can we harness the randomness of non-private bandit algorithms in the design private bandit algorithms? Our current privacy analysis ignores this additional source of randomness.

## Footnotes

[1] As defined, differential privacy requires indistinguishable outputs only for nonadaptively chosen sequences (that is, sequences where the inputs at time $t$ are fixed ahead of time and do not depend on the outputs at times $1, ..., t - 1$). The algorithms in our paper (and in previous work) in fact satisfy a stronger *adaptive* variant, in which an adversary selects the input online as the computation proceeds. When $\delta = 0$, the nonadaptive and adaptive variants are equivalent [DNPR10]. Moreover, protocols based on "randomized response" or the "tree-based sum" protocol of [DNPR10, CSS10] are adaptively secure, even when $\delta > 0$. We do not define the adaptive variant here explicitly, but we use it implicitly when proving privacy.

[2]Specifically, Dwork et al. [DNPR10] provide single-entry-level privacy, in the sense that a neighboring data set may only differ in one entry of the cost vector for one round. In contrast, we allow the entire cost vector to change at one round. Hiding that larger set of possible changes is more difficult, so our algorithms also satisfy the weaker notion of Dwork et al.

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
