[Supplementary Material 1]

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

# A  Algorithm for Tree based Aggregation Protocol

---

**Algorithm 2** Private Tree based aggregation protocol

---

**Input:** Vectors: $\langle z_1, \cdots, z_T \in \mathbb{R}^p \rangle$ (in an online sequence), $\mu$ : $L_2$-norm bound on $z_i$'s, privacy parameter: $\epsilon$.

1: **Initialization:** Define a binary tree of size $2^{\lceil \log_2 T \rceil + 1} - 1$ with leaves $z_1, \cdots, z_T$.
2: **Online Phase:** At each iteration $t \in [T]$, execute Steps 3 to 18.
3: Accept $z_t$ from the data stream.
4: Let $L = \{z_t \to \cdots \to root\}$ be the path from $z_t$ to the root.
5: **Tree update:** Steps 6 till 10.
6: $\Lambda \leftarrow$ First node in $L$ that is a left-child in $A$. Let $L_\Lambda = \{a_t \to \cdots \to \Lambda\}$.
7: **for all** $\alpha$ in $L$ **do**
8:    $\alpha \leftarrow \alpha + z_t$.
9:    **If** $\alpha \in L_\Lambda$, **then** $\alpha \leftarrow \alpha + n$, where $n \sim \lambda e^{-\frac{\|n\|_2 \epsilon}{\mu(\lceil \log_2 T \rceil + 1)}}$ and $\lambda$ is the proportionality constant.
10: **end for**
11: **Output private partial sum:** Steps 12 till 18.
12: Initialize vector $v \in \mathbb{R}^p$ to zero. Let $b \leftarrow \lceil \log_2 T \rceil + 1$-bit binary representation of $t$.
13: **for all** $i$ in $[\lceil \log_2 T \rceil + 1]$ **do**
14:    **if** bit $b_i = 1$ **then**
15:       **If** $i$-th node in $L$ (denoted by $L(i)$) is the left child in $A$, **then** $v \leftarrow v + L(i)$, **else** $v \leftarrow v + left\ sibling(L(i))$.
16:    **end if**
17: **end for**
18: **return** The noisy partial sum $v$.

---

# B  Privacy and Utility Guarantees of PFTAL Algorithm (Algorithm 1)

## B.1  Privacy guarantee for Algorithm 1

*Proof of Theorem 3.* Notice that given $\hat{v}_2, \cdots, \hat{v}_{t+1}$ (where $\hat{v}_{t+1}$ is the noisy version of $v_{t+1} = \sum_{\tau=1}^{t} \triangledown f_t(\hat{w}_\tau)$), the outputs $\hat{w}_2, \cdots \hat{w}_{t+1}$ are completely determined. Hence, it suffices to argue for the privacy of $\hat{v}_2, \cdots, \hat{v}_T$. Let $F$ and $F'$ be any two sequences of $L$-Lipschitz, $H$-strongly convex cost functions differing in exactly one cost function. Let $\hat{V} = \langle v_2, \cdots, v_T \rangle$. For $\epsilon$-differential privacy, we need to argue that for any set $S = \langle s_2, \cdots, s_T \rangle$ of $T$ vectors, the following is true.

$$\frac{\Pr[\hat{V}(F) = S]}{\Pr[\hat{V}(F') = S]} = \prod_{t=2}^{T} \frac{\Pr[\hat{v}_t(F) = s_t | \hat{v}_2 = s_2, \cdots, \hat{v}_{t-1} = s_{t-1}]}{\Pr[\hat{v}_t(F') = s_t | \hat{v}_2 = s_2, \cdots, \hat{v}_{t-1} = s_{t-1}]} \le e^\epsilon \qquad (10)$$

Now in (10), each $\hat{v}_t$ is computed using the tree $A$ (see Algorithm 2) and hence fixing the values of the nodes in the tree $A$ completely determines $V(F)$. Let $A(F) = \langle \alpha_1(F), \cdots \alpha_{(2^{\lceil lg T \rceil + 1} - 1)}(F) \rangle$ be the in-order tree traversal of $A(F)$. To prove (10), it suffices to prove that for all possible assignments $A = \langle \alpha_1, \cdots \rangle$ to the tree, the following holds.

$$\frac{\Pr[A(F) = A]}{\Pr[A(F') = A]} = \prod_{t=1}^{(2^{\lceil \log_2 T \rceil + 1} - 1)} \frac{\Pr[\alpha_t(F) = \alpha_t | \alpha_1(F) = \alpha_1, \cdots, \alpha_{t-1}(F) = \alpha_{t-1}]}{\Pr[\alpha_t(F') = \alpha_t | \alpha_1(F') = \alpha_1, \cdots, \alpha_{t-1}(F') = \alpha_{t-1}]} \le e^\epsilon \tag{11}$$

In the above ratio, changing one entry in the data set $F$ affects only $(\lceil \log_2 T \rceil + 1)$ terms in the product in (11). By the amount of noise added to each node of the tree, each of the ratio in the product of (11) is bounded by $e^{\left(\frac{\epsilon}{\lceil \log_2 T \rceil + 1}\right)}$. (See Line 9 in Algorithm 2). Here we have used the fact that for any vector $w \in \mathcal{C}$, $\|\triangledown f_t(w)\|_2$ is at most $L$ (by the Lipschitz property) and $\|\triangledown \tilde{f}_t(w)\|_2$ (in (3)) is at most $L + H\|\mathcal{C}\|_2$ (by the bound on the convex set $\mathcal{C}$).

Hence, we can conclude that in overall, Algorithm 1 is $\epsilon$-differentially private. $\qquad \square$

## B.2 Regret guarantee for Algorithm 1

*Proof of Theorem 4.* Recall that regret is given by the following expression

$$Regret(T) = \sum_{t=1}^{T} f_t(\hat{w}_t) - \min_{w \in \mathcal{C}} \sum_{t=1}^{T} f_t(w). \tag{12}$$

We will prove the required regret bound via the following three stage argument. We will first show in Lemma 7 that the regret in (12) is upper bounded by the regret for the cost functions $\tilde{f}_t$ (see (3) for notation). Next in Lemma 8, we show that the regret for $\tilde{f}_t$'s with respect to $\hat{w}_t$'s is not "too much" higher compared to the regret with $\tilde{w}_t$'s (see (4) for notation). Finally we bound the regret with respect to $\tilde{w}_t$'s.

**Lemma 7.** $\sum_{t=1}^{T} f_t(\hat{w}_t) - \min_{w \in \mathcal{C}} \sum_{t=1}^{T} f_t(w) \leq \sum_{t=1}^{T} \tilde{f}_t(\hat{w}_t) - \min_{w \in \mathcal{C}} \sum_{t=1}^{T} \tilde{f}_t(w).$

*Proof.* First notice that by definition, $f_t(\hat{w}_t) = \tilde{f}_t(\hat{w}_t)$. Also, notice that $\tilde{f}_t(w) \leq f_t(w)$ for all $w \in \mathbb{R}^p$. There fore, i) $\sum_{t=1}^{T} f_t(\hat{w}_t) = \sum_{t=1}^{T} \tilde{f}_t(\hat{w}_t)$ and ii) $\min_{w \in \mathcal{C}} \sum_{t=1}^{T} \tilde{f}_t(w) \leq \min_{w \in \mathcal{C}} \sum_{t=1}^{T} f_t(w)$.

This completes the proof. $\square$

In the next lemma we show that the regret with the outputs $\hat{w}_1, \cdots, \hat{w}_T$ is not much different from with respect to $\tilde{w}_1, \cdots, \tilde{w}_T$.

**Lemma 8.** *Under the randomness of Algorithm 1, the following is true.*

$$\mathbb{E}\left[\sum_{t=1}^{T} \tilde{f}_t(\hat{w}_t) - \min_{w \in \mathcal{C}} \sum_{t=1}^{T} \tilde{f}_t(w)\right] \leq \mathbb{E}\left[\sum_{t=1}^{T} \tilde{f}_t(\tilde{w}_t) - \min_{w \in \mathcal{C}} \sum_{t=1}^{T} \tilde{f}_t(w)\right] + \frac{4p(L + H\|\mathcal{C}\|_2)^2 \log^{2.5} T}{\epsilon H}.$$

*Proof.* Recall that $\tilde{w}_{t+1} = \arg\min_{w \in \mathcal{C}} J(w)$, where $J(w) = \langle \sum_{\tau=1}^{t} \nabla f_t(\hat{w}_\tau), w \rangle + \frac{H}{2} \sum_{\tau=1}^{t} \|w - \hat{w}_\tau\|_2^2$. We can equivalently write $\hat{w}_{t+1} = \arg\min_{w \in \mathcal{C}} J(w) + \langle n, w \rangle$, where $n$ is the noise added in the noisy computation of $v_{t+1} = \sum_{\tau=1}^{t} \nabla f_t(\hat{w}_\tau)$ in Line 9 (via the tree-aggregation scheme). By the $Ht$-strong convexity property of $J(w)$, we have

$$\|\tilde{w}_{t+1} - \hat{w}_{t+1}\|_2 \leq \frac{2\|n\|_2}{Ht}. \tag{13}$$

Now, since $f_t$ is assumed to be $L$-Lipschitz and the $L_2$ norm of any vector in $\mathcal{C}$ is bounded by $\|\mathcal{C}\|_2$, it directly follows that $\tilde{f}_t$ is $(L + H\|\mathcal{C}\|_2)$-Lipschitz. Therefore, from (13) and using the Lipschitz property of $\tilde{f}_t$, we have

$$|\tilde{f}_t(\hat{w}_t) - \tilde{f}_t(\tilde{w}_t)| \leq \frac{2\|n\|_2(L + H\|\mathcal{C}\|_2)}{Ht}.$$

Therefore,

$$\mathbb{E}\left[\sum_{t=1}^{T} \tilde{f}_t(\hat{w}_t) - \min_{w \in \mathcal{C}} \sum_{t=1}^{T} \tilde{f}_t(w)\right] \leq \sum_{t=1}^{T} \tilde{f}_t(\tilde{w}_t) - \min_{w \in \mathcal{C}} \sum_{t=1}^{T} \tilde{f}_t(w) + \frac{2\mathbb{E}[\|n\|_2](L + H\|\mathcal{C}\|_2)}{H} \sum_{t=1}^{T} \frac{1}{t}$$

$$\leq \sum_{t=1}^{T} \tilde{f}_t(\tilde{w}_t) - \min_{w \in \mathcal{C}} \sum_{t=1}^{T} \tilde{f}_t(w) + \frac{2\mathbb{E}[\|n\|_2](L + H\|\mathcal{C}\|_2) \log T}{H}.$$

$$\tag{14}$$

$$\tag{15}$$

To bound $\mathbb{E}[\|n\|_2]$, notice that $n$ is formed by adding at most $\lceil \log T \rceil + 1$ vectors whose norms are drawn from the Gamma distribution with scale $p$ and shape $\frac{(\lceil \log T \rceil + 1)(L + H\|\mathcal{C}\|_2)}{\epsilon}$. Therefore, $\mathbb{E}[\|n\|_2] \leq \frac{4p \log^{1.5} T(L + H\|\mathcal{C}\|_2)}{\epsilon}$.

Plugging in the above bound in (15), we complete the proof. $\square$

Next, we prove the following fact which will be useful in proving the regret bound. In the on-line learning literature this fact is also called the bound on regret via the bound on forward regret [HAK07].

**Fact 9.** $\sum_{t=1}^{T} \tilde{f}_t(\tilde{w}_t) - \min_{w \in \mathcal{C}} \sum_{t=1}^{T} \tilde{f}_t(w) \leq \sum_{t=1}^{T} \tilde{f}_t(\tilde{w}_t) - \tilde{f}_t(\tilde{w}_{t+1}).$

*Proof.* We prove the above fact by proving that $\sum_{t=1}^{T} \tilde{f}_t(\tilde{w}_{t+1}) \leq \min_{w \in \mathcal{C}} \sum_{t=1}^{T} \tilde{f}_t(w)$. We prove this by induction. Clearly the base case is true by definition of $\tilde{w}_2$ (see (4)). Now assume correctness for $T - 1$, and

$$\sum_{t=1}^{T} \tilde{f}_t(\tilde{w}_{t+1}) \leq \min_{w \in \mathcal{C}} \sum_{t=1}^{T-1} \tilde{f}_t(w) + \tilde{f}_T(w_{T+1}) \quad \text{(by induction hypothesis)}$$

$$\leq \sum_{t=1}^{T-1} \tilde{f}_t(w_{T+1}) + \tilde{f}_T(w_{T+1})$$

$$= \min_{w \in \mathcal{C}} \sum_{t=1}^{T} \tilde{f}_t(w) \quad \text{(by definition)}.$$

$\square$

Let $\zeta = \sum_{t=1}^{T} \tilde{f}_t(\tilde{w}_t) - \min_{w \in \mathcal{C}} \sum_{t=1}^{T} \tilde{f}_t(w)$. Using Fact 9 above and the Lipschitz property of $\tilde{f}_t$'s, we can conclude that $\zeta \leq (L + H\|\mathcal{C}\|_2) \sum_{t=1}^{T} \|\tilde{w}_t - \tilde{w}_{t+1}\|_2$. All we now need to do is bound $\|\tilde{w}_t - \tilde{w}_{t+1}\|_2$ for all $t$.

**Claim 10.** *For all $t$, $\|\tilde{w}_t - \tilde{w}_{t+1}\|_2 \leq \frac{2(L + H\|\mathcal{C}\|_2)}{Ht}$.*

*Proof.* Notice that

$$\tilde{w}_t = \arg\min_{w \in \mathcal{C}} \sum_{\tau=1}^{t-1} \tilde{f}_\tau(w)$$

and

$$\tilde{w}_{t+1} = \arg\min_{w \in \mathcal{C}} \sum_{\tau=1}^{t-1} \tilde{f}_\tau(w) + \tilde{f}_t(w).$$

Let $J(w) = \sum_{\tau=1}^{t-1} \tilde{f}_\tau(w) + f_t(w)$. Therefore,

$$J(\tilde{w}_t) \geq J(\tilde{w}_{t+1}) + \frac{Ht}{2}\|\tilde{w}_t - \tilde{w}_{t+1}\|_2^2$$

$$\Leftrightarrow \frac{Ht}{2}\|\tilde{w}_t - \tilde{w}_{t+1}\|_2^2 \leq \left(\sum_{\tau=1}^{t-1} \tilde{f}_\tau(\tilde{w}_t) - \sum_{\tau=1}^{t-1} \tilde{f}_\tau(\tilde{w}_{t+1})\right) + \tilde{f}_t(\tilde{w}_t) - \tilde{f}_t(\tilde{w}_{t+1})$$

$$\Leftrightarrow \frac{Ht}{2}\|\tilde{w}_t - \tilde{w}_{t+1}\|_2^2 \leq f_t(\tilde{w}_t) - f_t(\tilde{w}_{t+1}) \leq (L + H\|\mathcal{C}\|_2)\|\tilde{w}_t - \tilde{w}_{t+1}\|_2$$

$$\Leftrightarrow \|\tilde{w}_t - \tilde{w}_{t+1}\|_2 \leq \frac{2(L + H\|\mathcal{C}\|_2)}{Ht}.$$

$\square$

Using the above claim and Fact 9, we can conclude that

$$\sum_{t=1}^{T} \tilde{f}_t(\tilde{w}_t) - \tilde{f}_t(\tilde{w}_{t+1}) \leq \frac{2(L + H\|\mathcal{C}\|_2)^2 \log T}{H}.$$

Combining the above expression with Lemma 8, we obtain the required regret bound. $\qquad\square$

## C  Results for General Convex Costs

In this section we will adapt the Private Follow the Approximate Leader (Algorithm 1) for $H$-strongly convex costs from previous section to the to the case of general convex functions. The idea is to add a $L_2$-regularizer to the cost functions while running the PFTAL algorithm, and then tune $H$ for the optimal regularization parameter. To be more precise, for every cost function $f_t$, we will have Algorithm 1 work with the cost function $h_t(w) = f_t(w) + \frac{H}{2}\|w\|_2^2$ (instead of $f_t$). Clearly, each $h_t$ is now $H$-strongly convex. So, the privacy and regret guarantees in Section 2.1.1 will hold for the cost sequence $h_1, \cdots, h_T$. Notice that the following is always true for any sequence of vectors $w_1, \cdots, w_T \in \mathcal{C}$, since the diameter of the convex set $\mathcal{C}$ is bounded.

$$\sum_{t=1}^{T} f_t(w_t) - \min_{w \in \mathcal{C}} \sum_{t=1}^{T} f_t(w) \leq \left( \sum_{t=1}^{T} h_t(w_t) - \min_{w \in \mathcal{C}} \sum_{t=1}^{T} h_t(w) \right) + \frac{HT}{2}\|\mathcal{C}\|_2^2. \qquad (16)$$

If $\hat{w}_1, \cdots, \hat{w}_T$ be the sequence of outputs of Algorithm 1 on the cost sequence $h_1, \cdots, h_T$, then (17) follows from Theorem 4 and (16).

$$\mathbb{E}\left[ \sum_{t=1}^{T} f_t(\hat{w}_t) - \min_{w \in \mathcal{C}} \sum_{t=1}^{T} f_t(w) \right] \leq \mathbb{E}\left[ \sum_{t=1}^{T} h_t(\hat{w}_t) - \min_{w \in \mathcal{C}} \sum_{t=1}^{T} h_t(w) \right] + \frac{HT}{2}\|\mathcal{C}\|_2^2$$

$$= O\left( \frac{p(L + H\|\mathcal{C}\|_2)^2 \log^{2.5} T}{\epsilon H} \right) + \frac{HT}{2}\|\mathcal{C}\|_2^2. \qquad (17)$$

**Theorem 11** (Regret guarantee). *Let $f_1, \cdots, f_T$ be L-Lipschitz convex functions and let $\mathcal{C} \subseteq \mathbb{R}^p$ be a fixed convex set. Setting the parameter $H$ in the regularizer $\frac{H}{2}\|w\|_2^2$ optimally, we have the following regret bound.*

$$\mathbb{E}\left[ \sum_{t=1}^{T} f_t(\hat{w}_t) - \min_{w \in \mathcal{C}} \sum_{t=1}^{T} f_t(w) \right] = O\left( \frac{\sqrt{p \log^{2.5} T}(L + \sqrt{\frac{p \log^{2.5} T}{\epsilon T}}\|\mathcal{C}\|_2)^2}{\epsilon} \sqrt{T} \right). \qquad (18)$$

*The expectation is over the randomness of the algorithm and the adversary.*

*Proof.* Setting $H = \sqrt{\frac{p \log^{2.5} T}{\epsilon T}}$ in the right hand side of (17), we get the regret guarantee in (18) for the sequence of outputs $\hat{w}_1, \cdots, \hat{w}_T$. $\qquad\square$

Notice that the regret bound in (18) is a factor or $\sqrt{p \log^{2.5} T}/\epsilon$ worse than the non-private regret bound of $O(\sqrt{T})$, assuming other parameters to be constants and $T = \omega\left(\frac{p}{\epsilon}\right)$. The assumption on $T$ is benign, since if $T = O\left(\frac{p}{\epsilon}\right)$, then the regret guarantee in (18) will no longer be sublinear.

We believe it is unlikely that one can remove the explicit dependence on the dimensionality in the regret bound for general convex costs, while preserving differential privacy.

## D  Regret Guarantees for Follow The Approximate Leader (Bandit version)

**Theorem 12** (Regret guarantee). *Let $\mathbb{B}^p$ be a d-dimensional unit ball centered at the origin and $\mathcal{C}$ be a convex set such that $r\mathbb{B}^p \subseteq \mathcal{C} \subseteq R\mathbb{B}^p$ (where $0 < r < R$).*

- **Adaptive adversary:** *Setting $\beta = \frac{p^{2/3}}{T^{1/4}}$ and $\xi = \beta/r$, the expected regret is at most*

$$\tilde{O}\left(p^{2/3}T^{3/4}\left(BR + \left(1 + \frac{R}{r}\right)L + \frac{(H\|\mathcal{C}\|_2 + B)^2}{H}\right)\right).$$

- **Oblivious adversary:** *Setting $\beta = \frac{p^{2/3}}{T^{1/3}}$ and $\xi = \beta/r$, the expected regret is at most*

$$\tilde{O}\left(p^{2/3}T^{2/3}\left((1 + R/r)L + \frac{(H\|\mathcal{C}\|_2 + B)^2}{H}\right)\right).$$

*The expectation is over the randomness of the algorithm and the adversary.*

## D.1 Proof: Regret guarantee for Adaptive Adversary

*Proof.* We prove the regret bound in the following three stages: i) In Lemma 13, we show that the regret for the output sequence $\hat{w}_1, \cdots, \hat{w}_t$ with respect to the original cost functions $f_t$'s is not much higher compared to $\hat{f}_t$'s with parameter vectors $\tilde{w}_1, \cdots, \tilde{w}_T$ (defined in (8)), ii) We show in Lemma 14 that the regret of $\hat{f}_t$'s with the parameter vectors $\tilde{w}_t$'s is at most the regret of the cost functions $\hat{g}_t$'s with the same parameter vectors (defined in (6)). iii) In Lemma 15, we directly bound the regret on $\hat{g}_t$'s with parameter vectors $\tilde{w}_t$'s.

**Lemma 13.** *For any sequence of parameter vectors $\tilde{w}_1, \cdots, \tilde{w}_T$ from the convex set $(1 - \xi)\mathcal{C}$ and vectors $\hat{w}_1, \cdots, \hat{w}_T$ such that for all $t \in [T]$, $\hat{w}_t = \tilde{w}_t + \beta u_t$ (where $u_t$ is a uniform vector drawn from the unit sphere $\mathbb{S}^{p-1}$), the following is true.*

$$\sum_{t=1}^{T} f_t(\hat{w}_t) - \min_{w \in \mathcal{C}} \sum_{t=1}^{T} f_t(w) \le \sum_{t=1}^{T} \hat{f}_t(\tilde{w}_t) - \min_{w \in (1-\xi)\mathcal{C}} \sum_{t=1}^{T} \hat{f}_t(w) + 3\beta LT + \xi RLT$$

*Proof.* First notice that for any $w \in \mathcal{C}$, the following is true for any $t \in [T]$ by the Lipschitz property of $f_t$'s.

$$
\begin{aligned}
\left| f_t(w) - \hat{f}_t(w) \right| &= |f_t(w) - \mathbb{E}_{v \sim \mathbb{B}^p}[f_t(w + \beta v)]| \\
&= |\mathbb{E}_v[f_t(w) - f_t(w + \beta v)]| \\
&\le L\beta \cdot \mathbb{E}_v[\|v\|_2] \le \beta L
\end{aligned}
\tag{19}
$$

Now for any $w \in \mathcal{C}$, by the Lipschitz property of $f_t$, we can obtain the following bound $|f_t(w) - f_t((1 - \xi)w)| \le \xi LR$. This means that $\min_{w \in (1-\xi)\mathcal{C}} \sum_{t=1}^{T} f_t(w) \le \min_{w \in \mathcal{C}} \sum_{t=1}^{T} f_t(w) + \xi LRT$. Therefore, by (19) we directly have

$$\min_{w \in (1-\xi)\mathcal{C}} \sum_{t=1}^{T} \hat{f}_t(w) \le \min_{w \in \mathcal{C}} \sum_{t=1}^{T} f_t(w) + \beta LT + \xi RLT \tag{20}$$

By Lipschitz property of $f_t$, we have $|f_t(\hat{w}_t) - f_t(\tilde{w}_t)| \le \beta L$. Additionally, by (22) we have $|\hat{f}_t(\tilde{w}_t) - f_t(\tilde{w}_t)| \le \beta L$. Combining these two observations, we get

$$\sum_{t=1}^{T} f_t(\hat{w}_t) \le \sum_{t=1}^{T} \hat{f}_t(\tilde{w}_t) + 2\beta LT \tag{21}$$

Combining (20) and (21) we get the required error guarantee. $\qquad\square$

**Lemma 14.** *For any sequence of parameter vectors $\tilde{w}_1, \cdots, \tilde{w}_T$ from the convex set $(1 - \xi)\mathcal{C}$, the following is true.*

$$\sum_{t=1}^{T} \hat{f}_t(\tilde{w}_t) - \min_{w \in (1-\xi)\mathcal{C}} \sum_{t=1}^{T} \hat{f}_t(w) \le \sum_{t=1}^{T} \hat{g}_t(\tilde{w}_t) - \min_{w \in (1-\xi)\mathcal{C}} \sum_{t=1}^{T} \hat{g}_t(w)$$

*Proof.* First notice that by definition, $\hat{f}_t(\tilde{w}_t) = \hat{g}_t(\tilde{w}_t)$. Also, notice that $\hat{g}_t(w) \leq \hat{f}_t(w)$ for all $w \in \mathbb{R}^p$. There fore, i) $\sum_{t=1}^{T} \hat{f}_t(\tilde{w}_t) = \sum_{t=1}^{T} \hat{g}_t(\tilde{w}_t)$ and ii) $\min_{w \in (1-\xi)\mathcal{C}} \sum_{t=1}^{T} \hat{f}_t(w) \leq \min_{w \in (1-\xi)\mathcal{C}} \sum_{t=1}^{T} \hat{g}_t(w)$.
This completes the proof. $\square$

Using the above lemma we directly get (22) below. In order to obtain the final regret guarantee, we just need to bound $\sum_{t=1}^{T} \hat{g}_t(\tilde{w}_t) - \min_{w \in (1-\xi)\mathcal{C}} \sum_{t=1}^{T} \hat{g}_t(w)$ and appropriately set $\beta$ and $\xi$.

$$\sum_{t=1}^{T} f_t(\hat{w}_t) - \min_{w \in \mathcal{C}} \sum_{t=1}^{T} f_t(w) \leq \sum_{t=1}^{T} \hat{g}_t(\tilde{w}_t) - \min_{w \in (1-\xi)\mathcal{C}} \sum_{t=1}^{T} \hat{g}_t(w) + 3\beta LT + \xi RLT \qquad (22)$$

**Lemma 15.** $\mathbb{E}\left[ \sum_{t=1}^{T} \hat{g}_t(\tilde{w}_t) - \min_{w \in (1-\xi)\mathcal{C}} \sum_{t=1}^{T} \hat{g}_t(w) \right] \leq \frac{2(H\|\mathcal{C}\|_2 + pB/\beta)^2}{H} \log T + 2BR\sqrt{T}\frac{p}{\beta}$. *The expectation is over the random unit vectors* $u_1, \cdots, u_T$.

*Proof.* Since $\tilde{g}_t$'s are $H$-strongly convex functions and $\left( H\|\mathcal{C}\|_2 + \frac{p}{\beta}B \right)$-Lipschitz, from the regret analysis in Lemma 8 we directly have the following.

$$\sum_{t=1}^{T} \tilde{g}_t(\tilde{w}_t) - \min_{w \in (1-\xi)\mathcal{C}} \sum_{t=1}^{T} \tilde{g}_t(w) \leq \frac{2(H\|\mathcal{C}\|_2 + \frac{p}{\beta}B)^2}{H} \log T \qquad (23)$$

Let $w^* = \arg\min_{w \in (1-\xi)\mathcal{C}} \sum_{t=1}^{T} \hat{g}_t(w)$. Therefore by (23), we have (24).

$$\sum_{t=1}^{T} \tilde{g}_t(\tilde{w}_t) - \sum_{t=1}^{T} \tilde{g}_t(w^*) \leq \frac{2(H\|\mathcal{C}\|_2 + \frac{p}{\beta}B)^2}{H} \log T \qquad (24)$$

Notice that

$$\mathbb{E}\left[ \sum_{t=1}^{T} \hat{g}_t(\tilde{w}_t) - \min_{w \in (1-\xi)\mathcal{C}} \sum_{t=1}^{T} \hat{g}_t(w) \right] = \mathbb{E}\left[ \sum_{t=1}^{T} \hat{g}_t(\tilde{w}_t) - \sum_{t=1}^{T} \hat{g}_t(w^*) \right]$$

$$= \mathbb{E}\left[ \sum_{t=1}^{T} \tilde{g}_t(\tilde{w}_t) \right] - \mathbb{E}\left[ \sum_{t=1}^{T} \hat{g}_t(w^*) \right] \qquad (25)$$

The last inequality follows from the observation that $\mathbb{E}_{u_t}[\tilde{g}_t(w)] = \hat{g}_t(w)$ for all $w \in \mathcal{C}$. Let $\alpha_t = \triangledown \mathbb{E}_{v \sim \mathbb{B}^p}\left[ f_t(\tilde{w}_t + \beta v) \right] - \frac{p}{\beta} f_t(\tilde{w}_t + \beta u_t)u_t$. For any $w \in (1-\xi)\mathcal{C}$,

$$\left| \sum_{t=1}^{T} (\hat{g}_t(w) - \tilde{g}_t(w)) \right| = \left| \langle w, \sum_{t=1}^{T} \alpha_t \rangle \right| \leq R\| \sum_{t=1}^{T} \alpha_t \|_2$$

Now,

$$\mathbb{E}\left[ \| \sum_{t=1}^{T} \alpha_t \|_2 \right]^2 \leq \mathbb{E}\left[ \| \sum_{t=1}^{T} \alpha_t \|_2^2 \right]$$

$$= \sum_{t=1}^{T} \mathbb{E}[\|\alpha_t\|_2^2] + 2\sum_{t<t'} \mathbb{E}\left[ \alpha_t \alpha_{t'} \right] \leq 4T\frac{p^2}{\beta^2}B^2$$

The last inequality is true because $\mathbb{E}\left[ \alpha_t \alpha_{t'} \right] = 0$. Therefore,

$$E\left[ \min_{w \in (1-\xi)\mathcal{C}} \sum_{t=1}^{T} \hat{g}_t(w) \right] \geq \mathbb{E}\left[ \min_{w \in (1-\xi)\mathcal{C}} \sum_{t=1}^{T} \tilde{g}_t(w) \right] - \frac{2p}{\beta} BR\sqrt{T}$$

Using this bound in (25), we have

$$\mathbb{E}\left[\sum_{t=1}^{T} \hat{g}_t(\tilde{w}_t) - \min_{w \in (1-\xi)\mathcal{C}} \sum_{t=1}^{T} \hat{g}_t(w)\right] \leq \mathbb{E}\left[\sum_{t=1}^{T} \tilde{g}_t(\tilde{w}_t) - \min_{w \in (1-\xi)\mathcal{C}} \sum_{t=1}^{T} \tilde{g}_t(w)\right] + \frac{2p}{\beta} BR\sqrt{T}$$

Plugging in the bound from (24) completes the proof. □

Combining Lemmas 13, 14 and 15, we obtain the following.

$$\mathbb{E}\left[\sum_{t=1}^{T} f_t(\hat{w}_t) - \min_{w \in \mathcal{C}} \sum_{t=1}^{T} f_t(w)\right] \leq 3\beta LT + \xi RLT + \frac{2(H\|\mathcal{C}\|_2 + \frac{p}{\beta}B)^2}{H} \log T + \frac{2p}{\beta} BR\sqrt{T}$$

Setting, $\beta = \frac{p^{2/3}}{T^{1/4}}$ and $\xi = \frac{\beta}{r}$ gives the required regret bound. □

### D.2 Proof: Regret guarantee for Oblivious Adversary

*Proof.* The proof of this theorem is similar to the proof with adaptive adversary, except we will be prove a tighter bound corresponding to Lemma 15.

**Lemma 16.** $\mathbb{E}\left[\sum_{t=1}^{T} \hat{g}_t(\tilde{w}_t) - \min_{w \in (1-\xi)\mathcal{C}} \sum_{t=1}^{T} \hat{g}_t(w)\right] \leq \frac{2(H\|\mathcal{C}\|_2 + pB/\beta)^2}{H} \log T$. *The expectation is over the random unit vectors* $u_1, \cdots, u_T$.

*Proof.* Similar to the proof of Lemma 15, let $w^* = \arg \min_{w \in (1-\xi)\mathcal{C}} \sum_{t=1}^{T} \hat{g}_t(w)$. Notice that $\mathbb{E}_{u_t}[\tilde{g}_t(w)] = \hat{g}_t(w)$ for all $w \in \mathcal{C}$. Therefore,

$$\mathbb{E}\left[\sum_{t=1}^{T} \hat{g}_t(\tilde{w}_t) - \sum_{t=1}^{T} \hat{g}_t(w^*)\right] = \mathbb{E}\left[\sum_{t=1}^{T} \tilde{g}_t(\tilde{w}_t)\right] - \mathbb{E}\left[\sum_{t=1}^{T} \tilde{g}_t(w^*)\right]$$

$$= \mathbb{E}\left[\sum_{t=1}^{T} \tilde{g}_t(\tilde{w}_t) - \sum_{t=1}^{T} \tilde{g}_t(w^*)\right]$$

$$\leq \mathbb{E}\left[\sum_{t=1}^{T} \tilde{g}_t(\tilde{w}_t) - \min_{w \in (1-\xi)\mathcal{C}} \sum_{t=1}^{T} \tilde{g}_t(w)\right] \quad (26)$$

Now, using the bound from (23) in (26), we get the required regret bound. □

Combining Lemmas 13, 14 and 16, we obtain the following.

$$\mathbb{E}\left[\sum_{t=1}^{T} f_t(\hat{w}_t) - \min_{w \in \mathcal{C}} \sum_{t=1}^{T} f_t(w)\right] \leq 3\beta LT + \xi RLT + \frac{2(H\|\mathcal{C}\|_2 + \frac{p}{\beta}B)^2}{H} \log T$$

Setting, $\beta = \frac{p^{2/3}}{T^{1/3}}$ and $\xi = \frac{\beta}{r}$ gives the required regret bound. □

## E    Algorithm and Regret Guarantees for Private Follow The Approximate Leader
## (Bandit version)

### E.1    Private Follow The Approximate Leader (Bandit version) Algorithm

**Algorithm 3** Differentially Private Follow the Approximate Leader (PFTAL): Bandit Version

---

**Input:** Cost functions: $\langle f_1, \cdots, f_T \rangle$ (in an online sequence), strong convexity parameter: $H$, bound on the costs: $B$, convex set: $\mathcal{C} \subseteq \mathbb{R}^p$, scaling parameter: $\xi$, sampling radius: $\beta$, and privacy parameter: $\epsilon$.

1: $w_1^\dagger \leftarrow$ Any vector from $\mathcal{C}$. **Output** $w_1^\dagger$.
2: Sample $u_1$ uniformly from the sphere $\mathbb{S}^{p-1} = \{w \in \mathbb{R}^p : \|w\|_2 = 1\}$.
3: Pass $\frac{p}{\beta} f_1(w_1^\dagger + \beta u_1) u_1$, $L_2$-bound $\frac{pB}{\beta}$ and privacy parameter $\epsilon$ to the *tree based protocol* (Algorithm 2) and receive the current partial sum in $v^\dagger_1$.
4: **for** time steps $t \in \{1, \cdots, T-1\}$ **do**

5: $\quad w_{t+1}^\dagger = \arg \min_{w \in (1-\xi)\mathcal{C}} \langle v^\dagger_t, w \rangle + \frac{H}{2} \sum_{\tau=1}^t \|w - w_\tau^\dagger\|_2^2$. **Output** $\hat{w}_t$.

6: $\quad$ Sample $u_{t+1}$ uniformly from the sphere $\mathbb{S}^{p-1}$.
7: $\quad$ Pass $\frac{p}{\beta} f_{t+1}(w_{t+1}^\dagger + \beta u_{t+1}) u_{t+1}$, $L_2$-bound $\frac{pB}{\beta}$ and privacy parameter $\epsilon$ to *the tree based protocol* (Algorithm 2) and receive the current partial sum in $v^\dagger_{t+1}$.
8: **end for**

---

### E.2 Regret Analysis

*Proof of Theorem 6.* Corresponding to definitions of $\hat{g}_t$ and $\tilde{g}_t$'s in (6), (7), and (8) (in Section 3.1), we redefine them while using the Taylor expansion around $w_{t+1}^\dagger$.

$$\hat{g}_t(w) = \hat{f}_t(w_t^\dagger) + \langle \nabla \hat{f}_t(w_t^\dagger), w - w_t^\dagger \rangle + \frac{H}{2} \|w - w_t^\dagger\|_2^2 \tag{27}$$

$$\tilde{g}_t(w) = \hat{f}_t(w_t^\dagger) - \langle \nabla \hat{f}_t(w_t^\dagger), w_t^\dagger \rangle + \langle \frac{p}{\beta} f_t(w_t^\dagger + \beta u_t) u_t, w \rangle + \frac{H}{2} \|w - w_t^\dagger\|_2^2 \tag{28}$$

$$\tilde{w}_{t+1} = \arg \min_{w \in (1-\xi)\mathcal{C}} \sum_{\tau=1}^t \tilde{g}_\tau(w) \tag{29}$$

With the above equations in hand, we can rewrite the definition of $w_{t+1}^\dagger$ in (9) as follows. Here $n_t = v_t^\dagger - v_t$, where $v^\dagger_t$ and $v_t$ are as defined in Section 3.2.

$$w_{t+1}^\dagger = \arg \min_{w \in (1-\xi)\mathcal{C}} \sum_{\tau=1}^t \tilde{g}_\tau(w) + \langle n_t, w \rangle \tag{30}$$

Using a similar argument we used in Lemma 8, we get the following.

$$\sum_{t=1}^T \hat{g}_t(w_t^\dagger) - \min_{w \in (1-\xi)\mathcal{C}} \sum_{t=1}^T \hat{g}_t(w) \le \sum_{t=1}^T \hat{g}_t(\tilde{w}_t) - \min_{w \in (1-\xi)\mathcal{C}} \sum_{t=1}^T \hat{g}_t(w) + \frac{2(pB/\beta + H\|\mathcal{C}\|_2)}{H} \sum_{t=1}^T \frac{\|n_t\|_2}{t} \tag{31}$$

From (31) and using an expectation bound on $\|n_t\|_2$ similar to Lemma 8, we obtain the following.

$$\mathbb{E}_{n_1, \cdots, n_T} \left[ \sum_{t=1}^T \hat{g}_t(w_t^\dagger) - \min_{w \in (1-\xi)\mathcal{C}} \sum_{t=1}^T \hat{g}_t(w) \Big| u_1, \cdots, u_T \right]$$

$$\le \mathbb{E}_{n_1, \cdots, n_T} \left[ \sum_{t=1}^T \hat{g}_t(\tilde{w}_t) - \min_{w \in (1-\xi)\mathcal{C}} \sum_{t=1}^T \hat{g}_t(w) \Big| u_1, \cdots, u_T \right] + \frac{2p(pB/\beta + H\|\mathcal{C}\|_2)^2 \log^{2.5} T}{\beta \epsilon H} \tag{32}$$

Now,

$$\mathbb{E}_{n_1, \cdots, n_T, u_1, \cdots, u_T} \left[ \sum_{t=1}^T \hat{g}_t(\tilde{w}_t) - \min_{w \in (1-\xi)\mathcal{C}} \sum_{t=1}^T \hat{g}_t(w) \right]$$

$$= \mathbb{E}_{n_1, \cdots, n_T} \left[ \mathbb{E}_{u_1, \cdots, u_T} \left[ \sum_{t=1}^T \hat{g}_t(\tilde{w}_t) - \min_{w \in (1-\xi)\mathcal{C}} \sum_{t=1}^T \hat{g}_t(w) \Big| n_1, \cdots, n_T \right] \right] \tag{33}$$

If the adversary is adaptive, then by the same line of argument in Lemma 15, we have

$$
\mathbb{E}_{u_1,\cdots,u_T}\left[\sum_{t=1}^{T}\hat{g}_t(\tilde{w}_t) - \min_{w\in(1-\xi)\mathcal{C}}\sum_{t=1}^{T}\hat{g}_t(w)\,\middle|\,n_1,\cdots,n_T\right]
$$

$$
\leq \mathbb{E}_{u_1,\cdots,u_T}\left[\sum_{t=1}^{T}\tilde{g}_t(\tilde{w}_t) - \min_{w\in(1-\xi)\mathcal{C}}\sum_{t=1}^{T}\tilde{g}_t(w)\,\middle|\,n_1,\cdots,n_T\right] + \frac{2p}{\beta}BR\sqrt{T} \tag{34}
$$

If the adversary is oblivious, then by the same line of argument in Lemma 16, we have

$$
\mathbb{E}_{u_1,\cdots,u_T}\left[\sum_{t=1}^{T}\hat{g}_t(\tilde{w}_t) - \min_{w\in(1-\xi)\mathcal{C}}\sum_{t=1}^{T}\hat{g}_t(w)\,\middle|\,n_1,\cdots,n_T\right]
$$

$$
\leq \mathbb{E}_{u_1,\cdots,u_T}\left[\sum_{t=1}^{T}\tilde{g}_t(\tilde{w}_t) - \min_{w\in(1-\xi)\mathcal{C}}\sum_{t=1}^{T}\tilde{g}_t(w)\,\middle|\,n_1,\cdots,n_T\right] \tag{35}
$$

For the purpose of brevity, we combine (34) and (35) into one expression (36), where the term $\gamma$ equals $\frac{2d}{\beta}\sqrt{TRB}$ for adaptive adversary and zero for oblivious adversary. For the rest of the proof, we will set $\gamma$ according to the assumption about the adversary.

$$
\mathbb{E}_{u_1,\cdots,u_T}\left[\sum_{t=1}^{T}\hat{g}_t(\tilde{w}_t) - \min_{w\in(1-\xi)\mathcal{C}}\sum_{t=1}^{T}\hat{g}_t(w)\,\middle|\,n_1,\cdots,n_T\right]
$$

$$
\leq \mathbb{E}_{u_1,\cdots,u_T}\left[\sum_{t=1}^{T}\tilde{g}_t(\tilde{w}_t) - \min_{w\in(1-\xi)\mathcal{C}}\sum_{t=1}^{T}\tilde{g}_t(w)\,\middle|\,n_1,\cdots,n_T\right] + \gamma \tag{36}
$$

Plugging (36) back in (33), we get

$$
\mathbb{E}_{n_1,\cdots,n_T,u_1,\cdots,u_T}\left[\sum_{t=1}^{T}\hat{g}_t(\tilde{w}_t) - \min_{w\in(1-\xi)\mathcal{C}}\sum_{t=1}^{T}\hat{g}_t(w)\right]
$$

$$
\leq \mathbb{E}_{n_1,\cdots,n_T}\left[\mathbb{E}_{u_1,\cdots,u_T}\left[\sum_{t=1}^{T}\tilde{g}_t(\tilde{w}_t) - \min_{w\in(1-\xi)\mathcal{C}}\sum_{t=1}^{T}\tilde{g}_t(w)\,\middle|\,n_1,\cdots,n_T\right]\right] + \gamma
$$

$$
= \mathbb{E}_{n_1,\cdots,n_T,u_1,\cdots,u_T}\left[\sum_{t=1}^{T}\tilde{g}_t(\tilde{w}_t) - \min_{w\in(1-\xi)\mathcal{C}}\sum_{t=1}^{T}\tilde{g}_t(w)\right] + \gamma \tag{37}
$$

Combining (32) and (37), we have

$$
\mathbb{E}_{n_1,\cdots,n_T,u_1,\cdots,u_T}\left[\sum_{t=1}^{T}\hat{g}_t(w_t^{\dagger}) - \min_{w\in(1-\xi)\mathcal{C}}\sum_{t=1}^{T}\hat{g}_t(w)\right]
$$

$$
= \mathbb{E}_{u_1,\cdots,u_T}\left[\mathbb{E}_{n_1,\cdots,n_T}\left[\sum_{t=1}^{T}\hat{g}_t(w_t^{\dagger}) - \min_{w\in(1-\xi)\mathcal{C}}\sum_{t=1}^{T}\hat{g}_t(w)\,\middle|\,u_1,\cdots,u_T\right]\right]
$$

$$
\leq \mathbb{E}_{u_1,\cdots,u_T}\left[\mathbb{E}_{n_1,\cdots,n_T}\left[\sum_{t=1}^{T}\hat{g}_t(\tilde{w}_t) - \min_{w\in(1-\xi)\mathcal{C}}\sum_{t=1}^{T}\hat{g}_t(w)\,\middle|\,u_1,\cdots,u_T\right]\right] + \frac{2p(pB/\beta + H\|\mathcal{C}\|_2)^2\log^{2.5}T}{\beta\epsilon H}
$$

$$
= \mathbb{E}_{n_1,\cdots,n_T,u_1,\cdots,u_T}\left[\sum_{t=1}^{T}\hat{g}_t(\tilde{w}_t) - \min_{w\in(1-\xi)\mathcal{C}}\sum_{t=1}^{T}\hat{g}_t(w)\right] + \frac{2p(pB/\beta + H\|\mathcal{C}\|_2)^2\log^{2.5}T}{\beta\epsilon H}
$$

$$
\leq \mathbb{E}_{n_1,\cdots,n_T,u_1,\cdots,u_T}\left[\sum_{t=1}^{T}\tilde{g}_t(\tilde{w}_t) - \min_{w\in(1-\xi)\mathcal{C}}\sum_{t=1}^{T}\tilde{g}_t(w)\right] + \frac{2p(pB/\beta + H\|\mathcal{C}\|_2)^2\log^{2.5}T}{\beta\epsilon H} + \gamma
$$

$$
\tag{38}
$$

Plugging in the absolute bound on $\sum_{t=1}^{T} \tilde{g}_t(\tilde{w}_t) - \min_{w \in (1-\xi)\mathcal{C}} \sum_{t=1}^{T} \tilde{g}_t(w)$ from (23), we obtain the following.

$$\mathbb{E}_{n_1, \cdots, n_T, u_1, \cdots, u_T} \left[ \sum_{t=1}^{T} \hat{g}_t(w_t^\dagger) - \min_{w \in (1-\xi)\mathcal{C}} \sum_{t=1}^{T} \hat{g}_t(w) \right]$$

$$\leq \frac{2(H\|\mathcal{C}\|_2 + \frac{p}{\beta}B)^2}{H} \log T + \frac{2p(pB/\beta + H\|\mathcal{C}\|_2)^2 \log^{2.5} T}{\beta \epsilon H} + \gamma \qquad (39)$$

Combining Lemmas 13, 14 and (39), we obtain the following. The expectation is over the complete randomness of the private FTAL (bandit version).

$$\mathbb{E}\left[ \sum_{t=1}^{T} f_t(\hat{w}_t) - \min_{w \in \mathcal{C}} \sum_{t=1}^{T} f_t(w) \right]$$

$$\leq 3\beta LT + \xi RLT + \frac{2(H\|\mathcal{C}\|_2 + \frac{p}{\beta}B)^2}{H} \log T + \frac{2p(pB/\beta + H\|\mathcal{C}\|_2)^2 \log^{2.5} T}{\beta \epsilon H} + \gamma$$

Recall that if the adversary is adaptive, then $\gamma = \frac{2p}{\beta} BR\sqrt{T}$ and zero otherwise. Setting $\beta = \frac{p}{T^{1/4}}$ for adaptive adversary and $\beta = \frac{p}{T^{1/3}}$ for oblivious adversary, and setting $\xi = \frac{\beta}{r}$, we get the required regret bound. $\qquad \square$

### E.3 Private Bandit Learning for General Convex Functions

Our results in this section can be extended to the setting with general convex costs via the regularization "trick" from Appendix C (by adding $\frac{H}{2}\|w\|_2^2$ to each cost function $f_t$). One can show that under optimal choice of $H$, both for oblivious and adaptive adversary, the regret scales as $\tilde{O}(T^{3/4}/\epsilon)$, which is also the best known nonprivate bound [FKM05]. We provide the formal regret guarantee below.

**Theorem 17** (Regret guarantee)**.** *Let $\mathbb{B}^p$ be a $p$-dimensional unit ball centered at the origin and $\mathcal{C} \subseteq \mathbb{R}^p$ be a convex set such that $r\mathbb{B}^p \subseteq \mathcal{C} \subseteq R\mathbb{B}^p$ (where $0 < r < R$). Let $f_1, \cdots, f_T$ be $L$-Lipschitz functions and for all $w \in \mathcal{C}$, $|f_i(w)| \leq B$. Additionally assume that the regularizing parameter $H$ is set to $1/T^{1/4}$. Setting $\beta = \frac{p}{T^{1/4}}$ and $\xi = \beta/r$ in the Private Follow The Approximate Leader (bandit version) algorithm (Algorithm 3), we obtain the following regret guarantee.*

$$\mathbb{E}\left[ \sum_{t=1}^{T} f_t(\hat{w}_t) - \min_{w \in \mathcal{C}} \sum_{t=1}^{T} f_t(w) \right] \leq \tilde{O}\left( pT^{3/4}\chi \right).$$

*Here $\chi = \left( BR + (1 + R/r)L + \frac{B^3}{\epsilon} \right)$. The expectation is over the randomness of the algorithm and the adversary.*

[Supplementary Material 2 · Appendix.pdf]

# A  Algorithm for Tree based Aggregation Protocol

---
**Algorithm 2** Private Tree based aggregation protocol

---
**Input:** Vectors: $\langle z_1, \cdots, z_T \in \mathbb{R}^p \rangle$ (in an online sequence), $\mu$ : $L_2$-norm bound on $z_i$'s, privacy parameter: $\epsilon$.

1: **Initialization:** Define a binary tree of size $2^{\lceil \log_2 T \rceil + 1} - 1$ with leaves $z_1, \cdots, z_T$.
2: **Online Phase:** At each iteration $t \in [T]$, execute Steps 3 to 18.
3: Accept $z_t$ from the data stream.
4: Let $L = \{z_t \to \cdots \to root\}$ be the path from $z_t$ to the root.
5: **Tree update:** Steps 6 till 10.
6: $\Lambda \leftarrow$ First node in $L$ that is a left-child in $A$. Let $L_\Lambda = \{a_t \to \cdots \to \Lambda\}$.
7: **for all** $\alpha$ in $L$ **do**
8:     $\alpha \leftarrow \alpha + z_t$.
9:     **If** $\alpha \in L_\Lambda$, **then** $\alpha \leftarrow \alpha + n$, where $n \sim \lambda e^{-\frac{\|n\|_2 \epsilon}{\mu(\lceil \log_2 T \rceil + 1)}}$ and $\lambda$ is the proportionality constant.
10: **end for**
11: **Output private partial sum:** Steps 12 till 18.
12: Initialize vector $v \in \mathbb{R}^p$ to zero. Let $b \leftarrow \lceil \log_2 T \rceil + 1$-bit binary representation of $t$.
13: **for all** $i$ in $[\lceil \log_2 T \rceil + 1]$ **do**
14:     **if** bit $b_i = 1$ **then**
15:         **If** $i$-th node in $L$ (denoted by $L(i)$) is the left child in $A$, **then** $v \leftarrow v + L(i)$, **else** $v \leftarrow v + \text{left sibling}(L(i))$.
16:     **end if**
17: **end for**
18: **return** The noisy partial sum $v$.

---

# B  Privacy and Utility Guarantees of PFTAL Algorithm (Algorithm 1)

## B.1  Privacy guarantee for Algorithm 1

*Proof of Theorem 3.* Notice that given $\hat{v}_2, \cdots, \hat{v}_{t+1}$ (where $\hat{v}_{t+1}$ is the noisy version of $v_{t+1} = \sum_{\tau=1}^{t} \bigtriangledown f_t(\hat{w}_\tau)$), the outputs $\hat{w}_2, \cdots \hat{w}_{t+1}$ are completely determined. Hence, it suffices to argue for the privacy of $\hat{v}_2, \cdots, \hat{v}_T$. Let $F$ and $F'$ be any two sequences of $L$-Lipschitz, $H$-strongly convex cost functions differing in exactly one cost function. Let $\hat{V} = \langle v_2, \cdots, v_T \rangle$. For $\epsilon$-differential privacy, we need to argue that for any set $S = \langle s_2, \cdots, s_T \rangle$ of $T$ vectors, the following is true.

$$\frac{\Pr[\hat{V}(F) = S]}{\Pr[\hat{V}(F') = S]} = \prod_{t=2}^{T} \frac{\Pr[\hat{v}_t(F) = s_t | \hat{v}_2 = s_2, \cdots, \hat{v}_{t-1} = s_{t-1}]}{\Pr[\hat{v}_t(F') = s_t | \hat{v}_2 = s_2, \cdots, \hat{v}_{t-1} = s_{t-1}]} \leq e^\epsilon \quad (10)$$

Now in (10), each $\hat{v}_t$ is computed using the tree $A$ (see Algorithm 2) and hence fixing the values of the nodes in the tree $A$ completely determines $V(F)$. Let $A(F) = \langle \alpha_1(F), \cdots \alpha_{(2^{\lceil lg T \rceil + 1} - 1)}(F) \rangle$ be the in-order tree traversal of $A(F)$. To prove (10), it suffices to prove that for all possible assignments $A = \langle \alpha_1, \cdots \rangle$ to the tree, the following holds.

$$\frac{\Pr[A(F) = A]}{\Pr[A(F') = A]} = \prod_{t=1}^{(2^{\lceil \log_2 T \rceil + 1} - 1)} \frac{\Pr[\alpha_t(F) = \alpha_t | \alpha_1(F) = \alpha_1, \cdots, \alpha_{t-1}(F) = \alpha_{t-1}]}{\Pr[\alpha_t(F') = \alpha_t | \alpha_1(F') = \alpha_1, \cdots, \alpha_{t-1}(F') = \alpha_{t-1}]} \leq e^\epsilon \quad (11)$$

In the above ratio, changing one entry in the data set $F$ affects only $(\lceil \log_2 T \rceil + 1)$ terms in the product in (11). By the amount of noise added to each node of the tree, each of the ratio in the product of (11) is bounded by $e^{\left( \frac{\epsilon}{\lceil \log_2 T \rceil + 1} \right)}$. (See Line 9 in Algorithm 2). Here we have used the fact that for any vector $w \in \mathcal{C}$, $\| \bigtriangledown f_t(w) \|_2$ is at most $L$ (by the Lipschitz property) and $\| \bigtriangledown \tilde{f}_t(w) \|_2$ (in (3)) is at most $L + H \| \mathcal{C} \|_2$ (by the bound on the convex set $\mathcal{C}$).

Hence, we can conclude that in overall, Algorithm 1 is $\epsilon$-differentially private. $\square$

## B.2 Regret guarantee for Algorithm 1

*Proof of Theorem 4.* Recall that regret is given by the following expression

$$Regret(T) = \sum_{t=1}^{T} f_t(\hat{w}_t) - \min_{w \in \mathcal{C}} \sum_{t=1}^{T} f_t(w). \tag{12}$$

We will prove the required regret bound via the following three stage argument. We will first show in Lemma 7 that the regret in (12) is upper bounded by the regret for the cost functions $\tilde{f}_t$ (see (3) for notation). Next in Lemma 8, we show that the regret for $\tilde{f}_t$'s with respect to $\hat{w}_t$'s is not "too much" higher compared to the regret with $\tilde{w}_t$'s (see (4) for notation). Finally we bound the regret with respect to $\tilde{w}_t$'s.

**Lemma 7.** $\sum_{t=1}^{T} f_t(\hat{w}_t) - \min_{w \in \mathcal{C}} \sum_{t=1}^{T} f_t(w) \leq \sum_{t=1}^{T} \tilde{f}_t(\hat{w}_t) - \min_{w \in \mathcal{C}} \sum_{t=1}^{T} \tilde{f}_t(w).$

*Proof.* First notice that by definition, $f_t(\hat{w}_t) = \tilde{f}_t(\hat{w}_t)$. Also, notice that $\tilde{f}_t(w) \leq f_t(w)$ for all $w \in \mathbb{R}^p$. There fore, i) $\sum_{t=1}^{T} f_t(\hat{w}_t) = \sum_{t=1}^{T} \tilde{f}_t(\hat{w}_t)$ and ii) $\min_{w \in \mathcal{C}} \sum_{t=1}^{T} \tilde{f}_t(w) \leq \min_{w \in \mathcal{C}} \sum_{t=1}^{T} f_t(w).$

This completes the proof. $\qquad\qquad\qquad\qquad\qquad\qquad\qquad\qquad\qquad\qquad\qquad \square$

In the next lemma we show that the regret with the outputs $\hat{w}_1, \cdots, \hat{w}_T$ is not much different from with respect to $\tilde{w}_1, \cdots, \tilde{w}_T$.

**Lemma 8.** *Under the randomness of Algorithm 1, the following is true.*

$$\mathbb{E}\left[\sum_{t=1}^{T} \tilde{f}_t(\hat{w}_t) - \min_{w \in \mathcal{C}} \sum_{t=1}^{T} \tilde{f}_t(w)\right] \leq \mathbb{E}\left[\sum_{t=1}^{T} \tilde{f}_t(\tilde{w}_t) - \min_{w \in \mathcal{C}} \sum_{t=1}^{T} \tilde{f}_t(w)\right] + \frac{4p(L + H\|\mathcal{C}\|_2)^2 \log^{2.5} T}{\epsilon H}.$$

*Proof.* Recall that $\tilde{w}_{t+1} = \arg\min_{w \in \mathcal{C}} J(w)$, where $J(w) = \langle \sum_{\tau=1}^{t} \bigtriangledown f_t(\hat{w}_\tau), w \rangle + \frac{H}{2} \sum_{\tau=1}^{t} \|w - \hat{w}_\tau\|_2^2$. We can equivalently write $\hat{w}_{t+1} = \arg\min_{w \in \mathcal{C}} J(w) + \langle n, w \rangle$, where $n$ is the noise added in the noisy computation of $v_{t+1} = \sum_{\tau=1}^{t} \bigtriangledown f_t(\hat{w}_\tau)$ in Line 9 (via the tree-aggregation scheme). By the $Ht$-strong convexity property of $J(w)$, we have

$$\|\tilde{w}_{t+1} - \hat{w}_{t+1}\|_2 \leq \frac{2\|n\|_2}{Ht}. \tag{13}$$

Now, since $f_t$ is assumed to be $L$-Lipschitz and the $L_2$ norm of any vector in $\mathcal{C}$ is bounded by $\|\mathcal{C}\|_2$, it directly follows that $\tilde{f}_t$ is $(L + H\|\mathcal{C}\|_2)$-Lipschitz. Therefore, from (13) and using the Lipschitz property of $\tilde{f}_t$, we have

$$|\tilde{f}_t(\hat{w}_t) - \tilde{f}_t(\tilde{w}_t)| \leq \frac{2\|n\|_2(L + H\|\mathcal{C}\|_2)}{Ht}.$$

Therefore,

$$\mathbb{E}\left[\sum_{t=1}^{T} \tilde{f}_t(\hat{w}_t) - \min_{w \in \mathcal{C}} \sum_{t=1}^{T} \tilde{f}_t(w)\right] \leq \sum_{t=1}^{T} \tilde{f}_t(\tilde{w}_t) - \min_{w \in \mathcal{C}} \sum_{t=1}^{T} \tilde{f}_t(w) + \frac{2\mathbb{E}[\|n\|_2](L + H\|\mathcal{C}\|_2)}{H} \sum_{t=1}^{T} \frac{1}{t}$$

$$\leq \sum_{t=1}^{T} \tilde{f}_t(\tilde{w}_t) - \min_{w \in \mathcal{C}} \sum_{t=1}^{T} \tilde{f}_t(w) + \frac{2\mathbb{E}[\|n\|_2](L + H\|\mathcal{C}\|_2) \log T}{H}.$$

$$\tag{14}$$
$$\tag{15}$$

To bound $\mathbb{E}[\|n\|_2]$, notice that $n$ is formed by adding at most $\lceil \log T \rceil + 1$ vectors whose norms are drawn from the Gamma distribution with scale $p$ and shape $\frac{(\lceil \log T \rceil + 1)(L + H\|\mathcal{C}\|_2)}{\epsilon}$. Therefore, $\mathbb{E}[\|n\|_2] \leq \frac{4p \log^{1.5} T(L + H\|\mathcal{C}\|_2)}{\epsilon}$.

Plugging in the above bound in (15), we complete the proof. □

Next, we prove the following fact which will be useful in proving the regret bound. In the online learning literature this fact is also called the bound on regret via the bound on forward regret [HAK07].

**Fact 9.** $\sum_{t=1}^{T} \tilde{f}_t(\tilde{w}_t) - \min_{w \in \mathcal{C}} \sum_{t=1}^{T} \tilde{f}_t(w) \leq \sum_{t=1}^{T} \tilde{f}_t(\tilde{w}_t) - \tilde{f}_t(\tilde{w}_{t+1})$.

*Proof.* We prove the above fact by proving that $\sum_{t=1}^{T} \tilde{f}_t(\tilde{w}_{t+1}) \leq \min_{w \in \mathcal{C}} \sum_{t=1}^{T} \tilde{f}_t(w)$. We prove this by induction. Clearly the base case is true by definition of $\tilde{w}_2$ (see (4)). Now assume correctness for $T - 1$, and

$$\sum_{t=1}^{T} \tilde{f}_t(\tilde{w}_{t+1}) \leq \min_{w \in \mathcal{C}} \sum_{t=1}^{T-1} \tilde{f}_t(w) + \tilde{f}_T(w_{T+1}) \quad \text{(by induction hypothesis)}$$

$$\leq \sum_{t=1}^{T-1} \tilde{f}_t(w_{T+1}) + \tilde{f}_T(w_{T+1})$$

$$= \min_{w \in \mathcal{C}} \sum_{t=1}^{T} \tilde{f}_t(w) \quad \text{(by definition)}.$$

□

Let $\zeta = \sum_{t=1}^{T} \tilde{f}_t(\tilde{w}_t) - \min_{w \in \mathcal{C}} \sum_{t=1}^{T} \tilde{f}_t(w)$. Using Fact 9 above and the Lipschitz property of $\tilde{f}_t$'s, we can conclude that $\zeta \leq (L + H\|\mathcal{C}\|_2) \sum_{t=1}^{T} \|\tilde{w}_t - \tilde{w}_{t+1}\|_2$. All we now need to do is bound $\|\tilde{w}_t - \tilde{w}_{t+1}\|_2$ for all $t$.

**Claim 10.** *For all $t$,* $\|\tilde{w}_t - \tilde{w}_{t+1}\|_2 \leq \frac{2(L + H\|\mathcal{C}\|_2)}{Ht}$.

*Proof.* Notice that

$$\tilde{w}_t = \arg\min_{w \in \mathcal{C}} \sum_{\tau=1}^{t-1} \tilde{f}_\tau(w)$$

and

$$\tilde{w}_{t+1} = \arg\min_{w \in \mathcal{C}} \sum_{\tau=1}^{t-1} \tilde{f}_\tau(w) + \tilde{f}_t(w).$$

Let $J(w) = \sum_{\tau=1}^{t-1} \tilde{f}_\tau(w) + f_t(w)$. Therefore,

$$J(\tilde{w}_t) \geq J(\tilde{w}_{t+1}) + \frac{Ht}{2}\|\tilde{w}_t - \tilde{w}_{t+1}\|_2^2$$

$$\Leftrightarrow \frac{Ht}{2}\|\tilde{w}_t - \tilde{w}_{t+1}\|_2^2 \leq \left(\sum_{\tau=1}^{t-1} \tilde{f}_\tau(\tilde{w}_t) - \sum_{\tau=1}^{t-1} \tilde{f}_\tau(\tilde{w}_{t+1})\right) + \tilde{f}_t(\tilde{w}_t) - \tilde{f}_t(\tilde{w}_{t+1})$$

$$\Leftrightarrow \frac{Ht}{2}\|\tilde{w}_t - \tilde{w}_{t+1}\|_2^2 \leq f_t(\tilde{w}_t) - f_t(\tilde{w}_{t+1}) \leq (L + H\|\mathcal{C}\|_2)\|\tilde{w}_t - \tilde{w}_{t+1}\|_2$$

$$\Leftrightarrow \|\tilde{w}_t - \tilde{w}_{t+1}\|_2 \leq \frac{2(L + H\|\mathcal{C}\|_2)}{Ht}.$$

□

Using the above claim and Fact 9, we can conclude that

$$\sum_{t=1}^{T} \tilde{f}_t(\tilde{w}_t) - \tilde{f}_t(\tilde{w}_{t+1}) \leq \frac{2(L + H\|\mathcal{C}\|_2)^2 \log T}{H}.$$

Combining the above expression with Lemma 8, we obtain the required regret bound. □

## C   Results for General Convex Costs

In this section we will adapt the Private Follow the Approximate Leader (Algorithm 1) for $H$-strongly convex costs from previous section to the to the case of general convex functions. The idea is to add a $L_2$-regularizer to the cost functions while running the PFTAL algorithm, and then tune $H$ for the optimal regularization parameter. To be more precise, for every cost function $f_t$, we will have Algorithm 1 work with the cost function $h_t(w) = f_t(w) + \frac{H}{2}\|w\|_2^2$ (instead of $f_t$). Clearly, each $h_t$ is now $H$-strongly convex. So, the privacy and regret guarantees in Section 2.1.1 will hold for the cost sequence $h_1, \cdots, h_T$. Notice that the following is always true for any sequence of vectors $w_1, \cdots, w_T \in \mathcal{C}$, since the diameter of the convex set $\mathcal{C}$ is bounded.

$$\sum_{t=1}^{T} f_t(w_t) - \min_{w \in \mathcal{C}} \sum_{t=1}^{T} f_t(w) \leq \left( \sum_{t=1}^{T} h_t(w_t) - \min_{w \in \mathcal{C}} \sum_{t=1}^{T} h_t(w) \right) + \frac{HT}{2}\|\mathcal{C}\|_2^2. \tag{16}$$

If $\hat{w}_1, \cdots, \hat{w}_T$ be the sequence of outputs of Algorithm 1 on the cost sequence $h_1, \cdots, h_T$, then (17) follows from Theorem 4 and (16).

$$\mathbb{E}\left[ \sum_{t=1}^{T} f_t(\hat{w}_t) - \min_{w \in \mathcal{C}} \sum_{t=1}^{T} f_t(w) \right] \leq \mathbb{E}\left[ \sum_{t=1}^{T} h_t(\hat{w}_t) - \min_{w \in \mathcal{C}} \sum_{t=1}^{T} h_t(w) \right] + \frac{HT}{2}\|\mathcal{C}\|_2^2$$

$$= O\left( \frac{p(L + H\|\mathcal{C}\|_2)^2 \log^{2.5} T}{\epsilon H} \right) + \frac{HT}{2}\|\mathcal{C}\|_2^2. \tag{17}$$

**Theorem 11** (Regret guarantee). *Let $f_1, \cdots, f_T$ be L-Lipschitz convex functions and let $\mathcal{C} \subseteq \mathbb{R}^p$ be a fixed convex set. Setting the parameter $H$ in the regularizer $\frac{H}{2}\|w\|_2^2$ optimally, we have the following regret bound.*

$$\mathbb{E}\left[ \sum_{t=1}^{T} f_t(\hat{w}_t) - \min_{w \in \mathcal{C}} \sum_{t=1}^{T} f_t(w) \right] = O\left( \frac{\sqrt{p \log^{2.5} T}(L + \sqrt{\frac{p \log^{2.5} T}{\epsilon T}}\|\mathcal{C}\|_2)^2}{\epsilon} \sqrt{T} \right). \tag{18}$$

*The expectation is over the randomness of the algorithm and the adversary.*

*Proof.* Setting $H = \sqrt{\frac{p \log^{2.5} T}{\epsilon T}}$ in the right hand side of (17), we get the regret guarantee in (18) for the sequence of outputs $\hat{w}_1, \cdots, \hat{w}_T$. □

Notice that the regret bound in (18) is a factor or $\sqrt{p \log^{2.5} T}/\epsilon$ worse than the non-private regret bound of $O(\sqrt{T})$, assuming other parameters to be constants and $T = \omega\left(\frac{p}{\epsilon}\right)$. The assumption on $T$ is benign, since if $T = O\left(\frac{p}{\epsilon}\right)$, then the regret guarantee in (18) will no longer be sublinear.

We believe it is unlikely that one can remove the explicit dependence on the dimensionality in the regret bound for general convex costs, while preserving differential privacy.

## D   Regret Guarantees for Follow The Approximate Leader (Bandit version)

**Theorem 12** (Regret guarantee). *Let $\mathbb{B}^p$ be a d-dimensional unit ball centered at the origin and $\mathcal{C}$ be a convex set such that $r\mathbb{B}^p \subseteq \mathcal{C} \subseteq R\mathbb{B}^p$ (where $0 < r < R$).*

- **Adaptive adversary:** *Setting* $\beta = \frac{p^{2/3}}{T^{1/4}}$ *and* $\xi = \beta/r$ *, the expected regret is at most*

$$\tilde{O}\left(p^{2/3}T^{3/4}\left(BR + \left(1 + \frac{R}{r}\right)L + \frac{(H\|\mathcal{C}\|_2 + B)^2}{H}\right)\right).$$

- **Oblivious adversary:** *Setting* $\beta = \frac{p^{2/3}}{T^{1/3}}$ *and* $\xi = \beta/r$, *the expected regret is at most*

$$\tilde{O}\left(p^{2/3}T^{2/3}\left((1 + R/r)L + \frac{(H\|\mathcal{C}\|_2 + B)^2}{H}\right)\right).$$

*The expectation is over the randomness of the algorithm and the adversary.*

### D.1 Proof: Regret guarantee for Adaptive Adversary

*Proof.* We prove the regret bound in the following three stages: i) In Lemma 13, we show that the regret for the output sequence $\hat{w}_1, \cdots, \hat{w}_t$ with respect to the original cost functions $f_t$'s is not much higher compared to $\hat{f}_t$'s with parameter vectors $\tilde{w}_1, \cdots, \tilde{w}_T$ (defined in (8)), ii) We show in Lemma 14 that the regret of $\hat{f}_t$'s with the parameter vectors $\tilde{w}_t$'s is at most the regret of the cost functions $\hat{g}_t$'s with the same parameter vectors (defined in (6)). iii) In Lemma 15, we directly bound the regret on $\hat{g}_t$'s with parameter vectors $\tilde{w}_t$'s.

**Lemma 13.** *For any sequence of parameter vectors* $\tilde{w}_1, \cdots, \tilde{w}_T$ *from the convex set* $(1 - \xi)\mathcal{C}$ *and vectors* $\hat{w}_1, \cdots, \hat{w}_T$ *such that for all* $t \in [T]$, $\hat{w}_t = \tilde{w}_t + \beta u_t$ *(where* $u_t$ *is a uniform vector drawn from the unit sphere* $\mathbb{S}^{p-1}$*), the following is true.*

$$\sum_{t=1}^T f_t(\hat{w}_t) - \min_{w \in \mathcal{C}} \sum_{t=1}^T f_t(w) \le \sum_{t=1}^T \hat{f}_t(\tilde{w}_t) - \min_{w \in (1-\xi)\mathcal{C}} \sum_{t=1}^T \hat{f}_t(w) + 3\beta LT + \xi RLT$$

*Proof.* First notice that for any $w \in \mathcal{C}$, the following is true for any $t \in [T]$ by the Lipschitz property of $f_t$'s.

$$\left|f_t(w) - \hat{f}_t(w)\right| = |f_t(w) - \mathbb{E}_{v \sim \mathbb{B}^p}[f_t(w + \beta v)]|$$
$$= |\mathbb{E}_v[f_t(w) - f_t(w + \beta v)]|$$
$$\le L\beta \cdot \mathbb{E}_v[\|v\|_2] \le \beta L \tag{19}$$

Now for any $w \in \mathcal{C}$, by the Lipschitz property of $f_t$, we can obtain the following bound $|f_t(w) - f_t((1 - \xi)w)| \le \xi LR$. This means that $\min_{w \in (1-\xi)\mathcal{C}} \sum_{t=1}^T f_t(w) \le \min_{w \in \mathcal{C}} \sum_{t=1}^T f_t(w) + \xi LRT$. Therefore, by (19) we directly have

$$\min_{w \in (1-\xi)\mathcal{C}} \sum_{t=1}^T \hat{f}_t(w) \le \min_{w \in \mathcal{C}} \sum_{t=1}^T f_t(w) + \beta LT + \xi RLT \tag{20}$$

By Lipschitz property of $f_t$, we have $|f_t(\hat{w}_t) - f_t(\tilde{w}_t)| \le \beta L$. Additionally, by (22) we have $|\hat{f}_t(\tilde{w}_t) - f_t(\tilde{w}_t)| \le \beta L$. Combining these two observations, we get

$$\sum_{t=1}^T f_t(\hat{w}_t) \le \sum_{t=1}^T \hat{f}_t(\tilde{w}_t) + 2\beta LT \tag{21}$$

Combining (20) and (21) we get the required error guarantee. □

**Lemma 14.** *For any sequence of parameter vectors* $\tilde{w}_1, \cdots, \tilde{w}_T$ *from the convex set* $(1 - \xi)\mathcal{C}$, *the following is true.*

$$\sum_{t=1}^T \hat{f}_t(\tilde{w}_t) - \min_{w \in (1-\xi)\mathcal{C}} \sum_{t=1}^T \hat{f}_t(w) \le \sum_{t=1}^T \hat{g}_t(\tilde{w}_t) - \min_{w \in (1-\xi)\mathcal{C}} \sum_{t=1}^T \hat{g}_t(w)$$

*Proof.* First notice that by definition, $\hat{f}_t(\tilde{w}_t) = \hat{g}_t(\tilde{w}_t)$. Also, notice that $\hat{g}_t(w) \le \hat{f}_t(w)$ for all $w \in \mathbb{R}^p$. There fore, i) $\sum_{t=1}^{T} \hat{f}_t(\tilde{w}_t) = \sum_{t=1}^{T} \hat{g}_t(\tilde{w}_t)$ and ii) $\min_{w \in (1-\xi)\mathcal{C}} \sum_{t=1}^{T} \hat{f}_t(w) \le \min_{w \in (1-\xi)\mathcal{C}} \sum_{t=1}^{T} \hat{g}_t(w)$.

This completes the proof. $\square$

Using the above lemma we directly get (22) below. In order to obtain the final regret guarantee, we just need to bound $\sum_{t=1}^{T} \hat{g}_t(\tilde{w}_t) - \min_{w \in (1-\xi)\mathcal{C}} \sum_{t=1}^{T} \hat{g}_t(w)$ and appropriately set $\beta$ and $\xi$.

$$\sum_{t=1}^{T} f_t(\hat{w}_t) - \min_{w \in \mathcal{C}} \sum_{t=1}^{T} f_t(w) \le \sum_{t=1}^{T} \hat{g}_t(\tilde{w}_t) - \min_{w \in (1-\xi)\mathcal{C}} \sum_{t=1}^{T} \hat{g}_t(w) + 3\beta LT + \xi RLT \tag{22}$$

**Lemma 15.** $\mathbb{E}\left[\sum_{t=1}^{T} \hat{g}_t(\tilde{w}_t) - \min_{w \in (1-\xi)\mathcal{C}} \sum_{t=1}^{T} \hat{g}_t(w)\right] \le \frac{2(H\|\mathcal{C}\|_2 + pB/\beta)^2}{H} \log T + 2BR\sqrt{T}\frac{p}{\beta}$. *The expectation is over the random unit vectors $u_1, \cdots, u_T$.*

*Proof.* Since $\tilde{g}_t$'s are $H$-strongly convex functions and $\left(H\|\mathcal{C}\|_2 + \frac{p}{\beta}B\right)$-Lipschitz, from the regret analysis in Lemma 8 we directly have the following.

$$\sum_{t=1}^{T} \tilde{g}_t(\tilde{w}_t) - \min_{w \in (1-\xi)\mathcal{C}} \sum_{t=1}^{T} \tilde{g}_t(w) \le \frac{2(H\|\mathcal{C}\|_2 + \frac{p}{\beta}B)^2}{H} \log T \tag{23}$$

Let $w^* = \arg\min_{w \in (1-\xi)\mathcal{C}} \sum_{t=1}^{T} \hat{g}_t(w)$. Therefore by (23), we have (24).

$$\sum_{t=1}^{T} \tilde{g}_t(\tilde{w}_t) - \sum_{t=1}^{T} \tilde{g}_t(w^*) \le \frac{2(H\|\mathcal{C}\|_2 + \frac{p}{\beta}B)^2}{H} \log T \tag{24}$$

Notice that

$$\mathbb{E}\left[\sum_{t=1}^{T} \hat{g}_t(\tilde{w}_t) - \min_{w \in (1-\xi)\mathcal{C}} \sum_{t=1}^{T} \hat{g}_t(w)\right] = \mathbb{E}\left[\sum_{t=1}^{T} \hat{g}_t(\tilde{w}_t) - \sum_{t=1}^{T} \hat{g}_t(w^*)\right]$$

$$= \mathbb{E}[\sum_{t=1}^{T} \tilde{g}_t(\tilde{w}_t)] - \mathbb{E}[\sum_{t=1}^{T} \hat{g}_t(w^*)] \tag{25}$$

The last inequality follows from the observation that $\mathbb{E}_{u_t}[\tilde{g}_t(w)] = \hat{g}_t(w)$ for all $w \in \mathcal{C}$. Let $\alpha_t = \triangledown \mathbb{E}_{v \sim \mathbb{B}^p}\left[f_t(\tilde{w}_t + \beta v)\right] - \frac{p}{\beta} f_t(\tilde{w}_t + \beta u_t)u_t$. For any $w \in (1 - \xi)\mathcal{C}$,

$$\left|\sum_{t=1}^{T} (\hat{g}_t(w) - \tilde{g}_t(w))\right| = \left|\langle w, \sum_{t=1}^{T} \alpha_t \rangle\right| \le R\|\sum_{t=1}^{T} \alpha_t\|_2$$

Now,

$$\mathbb{E}\left[\|\sum_{t=1}^{T} \alpha_t\|_2\right]^2 \le \mathbb{E}\left[\|\sum_{t=1}^{T} \alpha_t\|_2^2\right]$$

$$= \sum_{t=1}^{T} \mathbb{E}[\|\alpha_t\|_2^2] + 2\sum_{t<t'} \mathbb{E}\left[\alpha_t \alpha_{t'}\right] \le 4T\frac{p^2}{\beta^2}B^2$$

The last inequality is true because $\mathbb{E}\left[\alpha_t \alpha_{t'}\right] = 0$. Therefore,

$$E\left[\min_{w \in (1-\xi)\mathcal{C}} \sum_{t=1}^{T} \hat{g}_t(w)\right] \ge \mathbb{E}\left[\min_{w \in (1-\xi)\mathcal{C}} \sum_{t=1}^{T} \tilde{g}_t(w)\right] - \frac{2p}{\beta}BR\sqrt{T}$$

Using this bound in (25), we have

$$\mathbb{E}\left[\sum_{t=1}^{T}\hat{g}_t(\tilde{w}_t) - \min_{w\in(1-\xi)\mathcal{C}}\sum_{t=1}^{T}\hat{g}_t(w)\right] \le \mathbb{E}\left[\sum_{t=1}^{T}\tilde{g}_t(\tilde{w}_t) - \min_{w\in(1-\xi)\mathcal{C}}\sum_{t=1}^{T}\tilde{g}_t(w)\right] + \frac{2p}{\beta}BR\sqrt{T}$$

Plugging in the bound from (24) completes the proof. □

Combining Lemmas 13, 14 and 15, we obtain the following.

$$\mathbb{E}\left[\sum_{t=1}^{T}f_t(\hat{w}_t) - \min_{w\in\mathcal{C}}\sum_{t=1}^{T}f_t(w)\right] \le 3\beta LT + \xi RLT + \frac{2(H\|\mathcal{C}\|_2 + \frac{p}{\beta}B)^2}{H}\log T + \frac{2p}{\beta}BR\sqrt{T}$$

Setting, $\beta = \frac{p^{2/3}}{T^{1/4}}$ and $\xi = \frac{\beta}{r}$ gives the required regret bound. □

### D.2   Proof: Regret guarantee for Oblivious Adversary

*Proof.* The proof of this theorem is similar to the proof with adaptive adversary, except we will be prove a tighter bound corresponding to Lemma 15.

**Lemma 16.** $\mathbb{E}\left[\sum_{t=1}^{T}\hat{g}_t(\tilde{w}_t) - \min_{w\in(1-\xi)\mathcal{C}}\sum_{t=1}^{T}\hat{g}_t(w)\right] \le \frac{2(H\|\mathcal{C}\|_2 + pB/\beta)^2}{H}\log T$. *The expectation is over the random unit vectors* $u_1, \cdots, u_T$.

*Proof.* Similar to the proof of Lemma 15, let $w^* = \arg\min_{w\in(1-\xi)\mathcal{C}}\sum_{t=1}^{T}\hat{g}_t(w)$. Notice that $\mathbb{E}_{u_t}[\tilde{g}_t(w)] = \hat{g}_t(w)$ for all $w \in \mathcal{C}$. Therefore,

$$\mathbb{E}\left[\sum_{t=1}^{T}\hat{g}_t(\tilde{w}_t) - \sum_{t=1}^{T}\hat{g}_t(w^*)\right] = \mathbb{E}\left[\sum_{t=1}^{T}\tilde{g}_t(\tilde{w}_t)\right] - \mathbb{E}\left[\sum_{t=1}^{T}\tilde{g}_t(w^*)\right]$$

$$= \mathbb{E}\left[\sum_{t=1}^{T}\tilde{g}_t(\tilde{w}_t) - \sum_{t=1}^{T}\tilde{g}_t(w^*)\right]$$

$$\le \mathbb{E}\left[\sum_{t=1}^{T}\tilde{g}_t(\tilde{w}_t) - \min_{w\in(1-\xi)\mathcal{C}}\sum_{t=1}^{T}\tilde{g}_t(w)\right] \quad (26)$$

Now, using the bound from (23) in (26), we get the required regret bound. □

Combining Lemmas 13, 14 and 16, we obtain the following.

$$\mathbb{E}\left[\sum_{t=1}^{T}f_t(\hat{w}_t) - \min_{w\in\mathcal{C}}\sum_{t=1}^{T}f_t(w)\right] \le 3\beta LT + \xi RLT + \frac{2(H\|\mathcal{C}\|_2 + \frac{p}{\beta}B)^2}{H}\log T$$

Setting, $\beta = \frac{p^{2/3}}{T^{1/3}}$ and $\xi = \frac{\beta}{r}$ gives the required regret bound. □

# E   Algorithm and Regret Guarantees for Private Follow The Approximate Leader
(Bandit version)

## E.1   Private Follow The Approximate Leader (Bandit version) Algorithm

**Algorithm 3** Differentially Private Follow the Approximate Leader (PFTAL): Bandit Version

---
**Input:** Cost functions: $\langle f_1, \cdots, f_T \rangle$ (in an online sequence), strong convexity parameter: $H$, bound on the costs: $B$, convex set: $\mathcal{C} \subseteq \mathbb{R}^p$, scaling parameter: $\xi$, sampling radius: $\beta$, and privacy parameter: $\epsilon$.

1: $w_1^\dagger \leftarrow$ Any vector from $\mathcal{C}$. **Output** $w_1^\dagger$.
2: Sample $u_1$ uniformly from the sphere $\mathbb{S}^{p-1} = \{w \in \mathbb{R}^p : \|w\|_2 = 1\}$.
3: Pass $\frac{p}{\beta} f_1(w_1^\dagger + \beta u_1)u_1$, $L_2$-bound $\frac{pB}{\beta}$ and privacy parameter $\epsilon$ to the *tree based protocol* (Algorithm 2) and receive the current partial sum in $v_1^\dagger$.
4: **for** time steps $t \in \{1, \cdots, T-1\}$ **do**
5:     $w_{t+1}^\dagger = \arg \min_{w \in (1-\xi)\mathcal{C}} \langle v_t^\dagger, w \rangle + \frac{H}{2} \sum_{\tau=1}^{t} \|w - w_\tau^\dagger\|_2^2$. **Output** $\hat{w}_t$.
6:     Sample $u_{t+1}$ uniformly from the sphere $\mathbb{S}^{p-1}$.
7:     Pass $\frac{p}{\beta} f_{t+1}(w_{t+1}^\dagger + \beta u_{t+1})u_{t+1}$, $L_2$-bound $\frac{pB}{\beta}$ and privacy parameter $\epsilon$ to *the tree based protocol* (Algorithm 2) and receive the current partial sum in $v_{t+1}^\dagger$.
8: **end for**

---

## E.2 Regret Analysis

*Proof of Theorem 6.* Corresponding to definitions of $\hat{g}_t$ and $\tilde{g}_t$'s in (6), (7), and (8) (in Section 3.1), we redefine them while using the Taylor expansion around $w_{t+1}^\dagger$.

$$\hat{g}_t(w) = \hat{f}_t(w_t^\dagger) + \langle \nabla \hat{f}_t(w_t^\dagger), w - w_t^\dagger \rangle + \frac{H}{2} \|w - w_t^\dagger\|_2^2 \tag{27}$$

$$\tilde{g}_t(w) = \hat{f}_t(w_t^\dagger) - \langle \nabla \hat{f}_t(w_t^\dagger), w_t^\dagger \rangle + \langle \frac{p}{\beta} f_t(w_t^\dagger + \beta u_t)u_t, w \rangle + \frac{H}{2} \|w - w_t^\dagger\|_2^2 \tag{28}$$

$$\tilde{w}_{t+1} = \arg \min_{w \in (1-\xi)\mathcal{C}} \sum_{\tau=1}^{t} \tilde{g}_\tau(w) \tag{29}$$

With the above equations in hand, we can rewrite the definition of $w_{t+1}^\dagger$ in (9) as follows. Here $n_t = v_t^\dagger - v_t$, where $v_t^\dagger$ and $v_t$ are as defined in Section 3.2.

$$w_{t+1}^\dagger = \arg \min_{w \in (1-\xi)\mathcal{C}} \sum_{\tau=1}^{t} \tilde{g}_\tau(w) + \langle n_t, w \rangle \tag{30}$$

Using a similar argument we used in Lemma 8, we get the following.

$$\sum_{t=1}^{T} \hat{g}_t(w_t^\dagger) - \min_{w \in (1-\xi)\mathcal{C}} \sum_{t=1}^{T} \hat{g}_t(w) \leq \sum_{t=1}^{T} \hat{g}_t(\tilde{w}_t) - \min_{w \in (1-\xi)\mathcal{C}} \sum_{t=1}^{T} \hat{g}_t(w) + \frac{2(pB/\beta + H\|\mathcal{C}\|_2)}{H} \sum_{t=1}^{T} \frac{\|n_t\|_2}{t} \tag{31}$$

From (31) and using an expectation bound on $\|n_t\|_2$ similar to Lemma 8, we obtain the following.

$$\mathbb{E}_{n_1, \cdots, n_T} \left[ \sum_{t=1}^{T} \hat{g}_t(w_t^\dagger) - \min_{w \in (1-\xi)\mathcal{C}} \sum_{t=1}^{T} \hat{g}_t(w) \middle| u_1, \cdots, u_T \right]$$

$$\leq \mathbb{E}_{n_1, \cdots, n_T} \left[ \sum_{t=1}^{T} \hat{g}_t(\tilde{w}_t) - \min_{w \in (1-\xi)\mathcal{C}} \sum_{t=1}^{T} \hat{g}_t(w) \middle| u_1, \cdots, u_T \right] + \frac{2p(pB/\beta + H\|\mathcal{C}\|_2)^2 \log^{2.5} T}{\beta \epsilon H} \tag{32}$$

Now,

$$\mathbb{E}_{n_1, \cdots, n_T, u_1, \cdots, u_T} \left[ \sum_{t=1}^{T} \hat{g}_t(\tilde{w}_t) - \min_{w \in (1-\xi)\mathcal{C}} \sum_{t=1}^{T} \hat{g}_t(w) \right]$$

$$= \mathbb{E}_{n_1, \cdots, n_T} \left[ \mathbb{E}_{u_1, \cdots, u_T} \left[ \sum_{t=1}^{T} \hat{g}_t(\tilde{w}_t) - \min_{w \in (1-\xi)\mathcal{C}} \sum_{t=1}^{T} \hat{g}_t(w) \middle| n_1, \cdots, n_T \right] \right] \tag{33}$$

If the adversary is adaptive, then by the same line of argument in Lemma 15, we have

$$
\mathbb{E}_{u_1,\cdots,u_T}\left[\sum_{t=1}^{T}\hat{g}_t(\tilde{w}_t) - \min_{w\in(1-\xi)\mathcal{C}}\sum_{t=1}^{T}\hat{g}_t(w)\,\middle|\,n_1,\cdots,n_T\right]
$$

$$
\leq \mathbb{E}_{u_1,\cdots,u_T}\left[\sum_{t=1}^{T}\tilde{g}_t(\tilde{w}_t) - \min_{w\in(1-\xi)\mathcal{C}}\sum_{t=1}^{T}\tilde{g}_t(w)\,\middle|\,n_1,\cdots,n_T\right] + \frac{2p}{\beta}BR\sqrt{T} \qquad (34)
$$

If the adversary is oblivious, then by the same line of argument in Lemma 16, we have

$$
\mathbb{E}_{u_1,\cdots,u_T}\left[\sum_{t=1}^{T}\hat{g}_t(\tilde{w}_t) - \min_{w\in(1-\xi)\mathcal{C}}\sum_{t=1}^{T}\hat{g}_t(w)\,\middle|\,n_1,\cdots,n_T\right]
$$

$$
\leq \mathbb{E}_{u_1,\cdots,u_T}\left[\sum_{t=1}^{T}\tilde{g}_t(\tilde{w}_t) - \min_{w\in(1-\xi)\mathcal{C}}\sum_{t=1}^{T}\tilde{g}_t(w)\,\middle|\,n_1,\cdots,n_T\right] \qquad (35)
$$

For the purpose of brevity, we combine (34) and (35) into one expression (36), where the term $\gamma$ equals $\frac{2d}{\beta}\sqrt{TRB}$ for adaptive adversary and zero for oblivious adversary. For the rest of the proof, we will set $\gamma$ according to the assumption about the adversary.

$$
\mathbb{E}_{u_1,\cdots,u_T}\left[\sum_{t=1}^{T}\hat{g}_t(\tilde{w}_t) - \min_{w\in(1-\xi)\mathcal{C}}\sum_{t=1}^{T}\hat{g}_t(w)\,\middle|\,n_1,\cdots,n_T\right]
$$

$$
\leq \mathbb{E}_{u_1,\cdots,u_T}\left[\sum_{t=1}^{T}\tilde{g}_t(\tilde{w}_t) - \min_{w\in(1-\xi)\mathcal{C}}\sum_{t=1}^{T}\tilde{g}_t(w)\,\middle|\,n_1,\cdots,n_T\right] + \gamma \qquad (36)
$$

Plugging (36) back in (33), we get

$$
\mathbb{E}_{n_1,\cdots,n_T,u_1,\cdots,u_T}\left[\sum_{t=1}^{T}\hat{g}_t(\tilde{w}_t) - \min_{w\in(1-\xi)\mathcal{C}}\sum_{t=1}^{T}\hat{g}_t(w)\right]
$$

$$
\leq \mathbb{E}_{n_1,\cdots,n_T}\left[\mathbb{E}_{u_1,\cdots,u_T}\left[\sum_{t=1}^{T}\tilde{g}_t(\tilde{w}_t) - \min_{w\in(1-\xi)\mathcal{C}}\sum_{t=1}^{T}\tilde{g}_t(w)\,\middle|\,n_1,\cdots,n_T\right]\right] + \gamma
$$

$$
= \mathbb{E}_{n_1,\cdots,n_T,u_1,\cdots,u_T}\left[\sum_{t=1}^{T}\tilde{g}_t(\tilde{w}_t) - \min_{w\in(1-\xi)\mathcal{C}}\sum_{t=1}^{T}\tilde{g}_t(w)\right] + \gamma \qquad (37)
$$

Combining (32) and (37), we have

$$
\mathbb{E}_{n_1,\cdots,n_T,u_1,\cdots,u_T}\left[\sum_{t=1}^{T}\hat{g}_t(w_t^\dagger) - \min_{w\in(1-\xi)\mathcal{C}}\sum_{t=1}^{T}\hat{g}_t(w)\right]
$$

$$
= \mathbb{E}_{u_1,\cdots,u_T}\left[\mathbb{E}_{n_1,\cdots,n_T}\left[\sum_{t=1}^{T}\hat{g}_t(w_t^\dagger) - \min_{w\in(1-\xi)\mathcal{C}}\sum_{t=1}^{T}\hat{g}_t(w)\,\middle|\,u_1,\cdots,u_T\right]\right]
$$

$$
\leq \mathbb{E}_{u_1,\cdots,u_T}\left[\mathbb{E}_{n_1,\cdots,n_T}\left[\sum_{t=1}^{T}\hat{g}_t(\tilde{w}_t) - \min_{w\in(1-\xi)\mathcal{C}}\sum_{t=1}^{T}\hat{g}_t(w)\,\middle|\,u_1,\cdots,u_T\right]\right] + \frac{2p(pB/\beta + H\|\mathcal{C}\|_2)^2\log^{2.5}T}{\beta\epsilon H}
$$

$$
= \mathbb{E}_{n_1,\cdots,n_T,u_1,\cdots,u_T}\left[\sum_{t=1}^{T}\hat{g}_t(\tilde{w}_t) - \min_{w\in(1-\xi)\mathcal{C}}\sum_{t=1}^{T}\hat{g}_t(w)\right] + \frac{2p(pB/\beta + H\|\mathcal{C}\|_2)^2\log^{2.5}T}{\beta\epsilon H}
$$

$$
\leq \mathbb{E}_{n_1,\cdots,n_T,u_1,\cdots,u_T}\left[\sum_{t=1}^{T}\tilde{g}_t(\tilde{w}_t) - \min_{w\in(1-\xi)\mathcal{C}}\sum_{t=1}^{T}\tilde{g}_t(w)\right] + \frac{2p(pB/\beta + H\|\mathcal{C}\|_2)^2\log^{2.5}T}{\beta\epsilon H} + \gamma
$$

$$
(38)
$$

Plugging in the absolute bound on $\sum_{t=1}^{T} \tilde{g}_t(\tilde{w}_t) - \min_{w \in (1-\xi)\mathcal{C}} \sum_{t=1}^{T} \tilde{g}_t(w)$ from (23), we obtain the following.

$$\mathbb{E}_{n_1,\cdots,n_T,u_1,\cdots,u_T}\left[\sum_{t=1}^{T} \hat{g}_t(w_t^{\dagger}) - \min_{w \in (1-\xi)\mathcal{C}} \sum_{t=1}^{T} \hat{g}_t(w)\right]$$

$$\leq \frac{2(H\|\mathcal{C}\|_2 + \frac{p}{\beta}B)^2}{H} \log T + \frac{2p(pB/\beta + H\|\mathcal{C}\|_2)^2 \log^{2.5} T}{\beta \epsilon H} + \gamma \qquad (39)$$

Combining Lemmas 13, 14 and (39), we obtain the following. The expectation is over the complete randomness of the private FTAL (bandit version).

$$\mathbb{E}\left[\sum_{t=1}^{T} f_t(\hat{w}_t) - \min_{w \in \mathcal{C}} \sum_{t=1}^{T} f_t(w)\right]$$

$$\leq 3\beta LT + \xi RLT + \frac{2(H\|\mathcal{C}\|_2 + \frac{p}{\beta}B)^2}{H} \log T + \frac{2p(pB/\beta + H\|\mathcal{C}\|_2)^2 \log^{2.5} T}{\beta \epsilon H} + \gamma$$

Recall that if the adversary is adaptive, then $\gamma = \frac{2p}{\beta}BR\sqrt{T}$ and zero otherwise. Setting $\beta = \frac{p}{T^{1/4}}$ for adaptive adversary and $\beta = \frac{p}{T^{1/3}}$ for oblivious adversary, and setting $\xi = \frac{\beta}{r}$, we get the required regret bound. $\qquad \square$

### E.3 Private Bandit Learning for General Convex Functions

Our results in this section can be extended to the setting with general convex costs via the regularization "trick" from Appendix C (by adding $\frac{H}{2}\|w\|_2^2$ to each cost function $f_t$). One can show that under optimal choice of $H$, both for oblivious and adaptive adversary, the regret scales as $\tilde{O}(T^{3/4}/\epsilon)$, which is also the best known nonprivate bound [FKM05]. We provide the formal regret guarantee below.

**Theorem 17** (Regret guarantee). *Let $\mathbb{B}^p$ be a $p$-dimensional unit ball centered at the origin and $\mathcal{C} \subseteq \mathbb{R}^p$ be a convex set such that $r\mathbb{B}^p \subseteq \mathcal{C} \subseteq R\mathbb{B}^p$ (where $0 < r < R$). Let $f_1, \cdots, f_T$ be $L$-Lipschitz functions and for all $w \in \mathcal{C}$, $|f_i(w)| \leq B$. Additionally assume that the regularizing parameter $H$ is set to $1/T^{1/4}$. Setting $\beta = \frac{p}{T^{1/4}}$ and $\xi = \beta/r$ in the Private Follow The Approximate Leader (bandit version) algorithm (Algorithm 3), we obtain the following regret guarantee.*

$$\mathbb{E}\left[\sum_{t=1}^{T} f_t(\hat{w}_t) - \min_{w \in \mathcal{C}} \sum_{t=1}^{T} f_t(w)\right] \leq \tilde{O}\left(pT^{3/4}\chi\right).$$

*Here $\chi = \left(BR + (1 + R/r)L + \frac{B^3}{\epsilon}\right)$. The expectation is over the randomness of the algorithm and the adversary.*