[Reviews · NeurIPS 2013]

Submitted by Assigned_Reviewer_1

This paper provides a generic way to perform parameter tuning of a given training algorithm in a differentially private manner.
Given a set of examples (for training and validation), a set of model parameters, a training algorithm, and a performance measure, the proposed procedure outputs a differentially private hypothesis with respect to prescribed privacy parameters.
The basic idea behind the procedure is the definition of (\beta_1, \beta_2, \delta)-stability; which describes the stability of the performance with respect to change in the training set and the validation set.
The procedure basically follows the exponential mechanism, and the utility bound is also provided.

At each step, the gradient is set to a tree such that each node of the tree represents the differentially private partial sum of values held by the descendent nodes. In execution of the differentially private FTAL, the learner issues a query to the tree to learn a differentially private parietal sum of previous gradients. Noting the fact that only O(log T) accesses are needed to learn the partial sum, the variance of the noise added to each value is O(log T).

The authors provides two algorithms, one is for the full information model and the other is for the bandit setting. In the full information setting, the proposed algorithm improves the regret bounds. The regret bound in the full information model with strongly convex and L-Lipschitz cost functions is significant; O(poly log T) is achieved as FTAL does. In the bandit setting, the authors introduces a technique called "one-shot gradient" to evaluate the gradient and shows regret bounds in several settings; the regret bound is optimal with respect to T when the cost function is strongly convex and L-Lipschitz.

The clarity can be improved. It is hard to follows the problem settings. It is not clearly stated what are private instances to be protected by differential privacy. Definitions of the oblivious adversary and the adaptive adversary in the bandit setting is unclear, too.

line 107. What is "stronger setting"? Does this mean "with stronger assumptions"?
line 229-232. The sentence was hard to follow.
line 305. This technique was referred to as one-point gradient, not as one-shot gradient.


Summary: Significant improvement of regret bounds of differentially private online learning in the full information model is shown.


Submitted by Assigned_Reviewer_7

This work provides new algorithms for differentially private online learning in basically all settings where results are known in the non-private case. The results very closely parallel (with some additional small factors and dependences) the best non-private bounds. There has been significant recent interest in differentially private learning, and this paper gives a much more comprehensive and compelling suite of results than previous work.

The approach is actually quite simple at its heart---do follow the approximate leader using a private (noisy) history that you maintain using standard algorithms for maintaining a differentially private counter. The natural suitability of follow the leader algorithms for privacy has been observed before, but the treatment here and the actual bounds obtained are very nice.

I'd prefer that you describe this work as giving a class of private learning algorithms, or a technique for constructing such algorithms, rather than a "general technique for making online algorithms differentially private" (since it's not the case that you can take an arbitrary online algorithm and make it private using this technique---right?).

The practice of citing references without listing authors made this paper hard to read---I was constantly flipping to the bibliography, since in many cases knowing the reference was important to understanding the ideas (e.g. in cases where the text did not make sufficiently clear whether the referenced work was in the private or non-private case).

In Table 1, it's probably worth reminding the reader that the delta in the first column comes from (eps, delta)-DP.

Why aren't there citations for the non-private results in Table 2? And also, the caption says it's the full-information setting, but the results are clearly for the bandit setting. And it seems your results for the adaptive case should have a T^3/4, not T^2/3, no?

Do you think the explicit dependences you see on the dimensionality are necessary when preserving privacy? Do you think there's a provable (small) gap between what is achievable in the dependence on T between private and non-private algorithms (accounting for your different poly-log factors)?

The second to last paragraph on p.6 has strange redundancy.


Summary: A simple and appealing approach and set of results on private online learning.

Submitted by Assigned_Reviewer_8

The paper presents a general method to design differentially private online algorithms, for the full information and bandit settings.

The paper is clear and well motivated. The relation with the relevant literature is thoroughly discussed.
I did not check all the details in the appendix, but the results appear correct to me.

I think the paper is probably not a breakthrough, but the ideas presented are interesting and I think that it still deserves to be published.

Minor things:
- please move the discussion to the main text, I find that the open problems are actually interesting, especially the comparison with the results in [1].
Summary: A general method to design differentially private online algorithms, for the full information and bandit settings. Clear and complete theoretical results are shown.
Author Feedback

Author rebuttal: Dear reviewers,
Thank you for your kind and detailed reviews. We will try to implement suggestions as best possible. Following are the specific responses to questions and comments.

We have copied the reviewers’ specific comments below, preceded by email quote character (>).

===============
Reviewer 1:
===============

1)
>It is not clearly stated what are private instances to be protected by differential privacy.

The private data set is the sequence of cost functions f_1, …, f_T. Each f_t can be thought as private information belonging to an individual. The output of the algorithm is the entire sequence of estimates $\hat w_1,...,\hat w_T$. Our notion of differential privacy requires that for every two sequences of cost functions that different in one cost function, the corresponding (distributions on) vectors of outputs be indistinguishable.

We have clarified this immediately before Definition 2.

2)
>Definitions of the oblivious adversary and the adaptive adversary in the bandit setting is unclear, too.

We clarified these on page two.

3)
>line 107. What is "stronger setting"? Does this mean "with stronger assumptions"?

The paper of Dwork et al. uses a weaker model for privacy than ours. The phrase “stronger setting” refers to the more stringent notion of privacy our algorithms provide.

Specifically, Dwork et al. provide “single-entry”-level privacy, in the sense that a neighboring data set may only differ in one entry of the cost vector for one round. In contrast, we allow the entire cost vector to change at one round. Hiding that larger set of possible changes is more difficult, so our algorithms satisfy the weaker notion of Dwork et al. a fortiori.

We clarified the difference in definitions in a footnote.

4)
>line 229-232. The sentence was hard to follow.

The definition of $\tilde w_{t}$ Equation (3) is equivalent to defining it as the minimizer of the sum of $\tilde f_i$; one obtains the expression in equation 3 by subtracting of the leading constant $f_t(\hat w_t)$ from each of the functions $\tilde f_t$.

We provided a clearer explanation.

5)
> line 305. This technique was referred to as one-point gradient, not as one-shot gradient.

We have seen both “one-shot” and “one-point” used in the literature (for example, one-shot is used in Shalev-Shwartz’s survey).

We addressed this issue by calling it one-point gradient throughout the paper.

===============
Reviewer 7:
===============
1)
> I'd prefer that you describe this work as giving a class of private learning algorithms, or a technique for constructing such algorithms, rather than a "general technique for making online algorithms differentially private" (since it's not the case that you can take an arbitrary online algorithm and make it private using this technique---right?).

Yes, you are right that we provide a class of private learning algorithm as opposed to a generic differentially private transformation for any online learning algorithm.

We clarified this in our presentation.

2)
> The practice of citing references without listing authors made this paper hard to read---I was constantly flipping to the bibliography, since in many cases knowing the reference was important to understanding the ideas (e.g. in cases where the text did not make sufficiently clear whether the referenced work was in the private or non-private case).

We fixed the bibliography issue.

3)
> In Table 1, it's probably worth reminding the reader that the delta in the first column comes from (eps, delta)-DP.

We introduced the (\eps,\delta)-variant of the differential privacy to take care of this issue.

3)
> Why aren't there citations for the non-private results in Table 2? And also, the caption says it's the full-information setting, but the results are clearly for the bandit setting. And it seems your results for the adaptive case should have a T^3/4, not T^2/3, no?

We addressed these issues.

4)
> Do you think the explicit dependence you see on the dimensionality are necessary when preserving privacy? Do you think there's a provable (small) gap between what is achievable in the dependence on T between private and non-private algorithms (accounting for your different poly-log factors)?

It is a very good question. We are not sure if the explicit dependence on dimensionality is necessary. In fact, in a recent result (not mentioned in this paper) by one of us one can show that if the cost functions are linear, then optimal dependence of log p can be obtained. It is unclear whether that result can be generalized.

5)
> The second to last paragraph on p.6 has strange redundancy.

We tried addressing the issue.

===============
Reviewer 8:
===============
1)
> Please move the discussion to the main text, I find that the open problems are actually interesting, especially the comparison with the results in [1].

We moved the discussion section to the main body.